# Rare genetic associations with human lifespan in UK Biobank are enriched for oncogenic genes

Junyoung Park [1] ✉, Andrés Peña-Tauber [1], Lia Talozzi[1], Michael D. Greicius[1,3] & Yann Le Guen [2,3]

Human lifespan is shaped by genetic and environmental factors. To enable precision health, understanding how genetic variants influence mortality is essential. We conducted a survival analysis in European ancestry participants of the UK Biobank, using age-at-death (N=35,551) and last-known-age (N=358,282). The associations identified were predominantly driven by cancer. We found lifespan-associated loci (*APOE*, *ZSCAN23*) for common variants and six genes where burden of loss-of-function variants were linked to reduced lifespan (*TET2*, *ATM*, *BRCA2*, *CKMT1B*, *BRCA1*, *ASXL1*). Additionally, eight genes with pathogenic missense variants were associated with reduced lifespan (*DNMT3A*, *SF3B1*, *TET2*, *PTEN*, *SOX21*, *TP53*, *SRSF2*, *RLIM*). Many of these genes are involved in oncogenic pathways and clonal hematopoiesis. Our findings highlight the importance of understanding genetic factors driving the most prevalent causes of mortality at a population level, highlighting the potential of early genetic testing to identify germline and somatic variants increasing one's susceptibility to cancer and/or early death.

Human lifespan is a complex trait influenced by both genetic and environmental factors and their interactions[1]. According to previous studies, genetics accounts for less than 10%[2] or up to 25% of the heritability of longevity[3]. Identifying the genetic variants that contribute to earlier death or prolonged survival can highlight key biological pathways linked to lifespan and inform genetic testing for general health and screening and enabling precision health. Previous genome-wide association studies (GWAS) have identified over 20 associated loci including *APOE*[4,5], *CHRNA3/5*[6], *HLA-DQA1* and *LPA*[7]. Recently, a burden analysis of protein-truncating variants from whole-exome sequencing (WES) data identified four additional genes (*BRCA2*, *BRCA1*, *ATM*, and *TET2*) linked to reduced lifespan[8]. However, most previous research on lifespan genetics has predominantly used proxy data, such as parents' age at death, due to a lack of proband lifespan data. While proxy-based GWAS have been useful to gain genomic insights into age-related diseases in cohorts primarily composed of middle-aged individuals, and show some consistency with associations related to lifespan[8], they may fail to fully capture the genetic influences that directly impact individual lifespan, particularly CHIP-related somatic variants[9]. On the other hand, some studies have employed logistic regression models on cases of extreme longevity and younger controls[10–12]. This approach may offer new insights by focusing on exceptionally long-lived individuals, yet they can be limited and costly. Moreover, replication of borderline significant variants remains an issue due to varying case definitions across studies, with some defining cases as individuals who survive to ages beyond 90 or 100 years or using the 90th or 99th survival percentiles as the age cutoff.

In this study, we carried out a genetic analysis of direct mortality data in the UK Biobank (UKB), the genetic database with the largest number of reported deaths (35,551 subjects) and aged individuals (344,237 subjects over 60 years old). To assess the association of genetic variants with lifespan in a survival analysis, we performed GWAS of common variants imputed from microarray data as well as

[1]Department of Neurology and Neurological Sciences, Stanford University, Stanford, CA 94305, USA. [2]Quantitative Sciences Unit, Department of Medicine, Stanford University, Stanford, CA 94304, USA. [3]These authors contributed equally: Michael D. Greicius, Yann Le Guen. ✉e-mail: jpark01@stanford.edu

burden/sequence kernel association test-optimized (SKAT-O) association of rare non-synonymous variants from WES data.

# Results

## Genome-wide association analyses in imputed array data

Our GWAS assessed 10,104,569 common variants (minor allele frequency (MAF) ≥ 0.1%) using Martingale residuals on 393,833 individuals including 35,551 deceased subjects (mean age at death: 71.2 years) and 358,282 living subjects (mean current age: 70.7) from UKB (Supplementary Table 1)[13]. Two loci reached genome-wide significance (GWS) ($p < 5.0 \times 10^{-8}$) on chromosomes 19 and 6 (Fig. 1A). On chromosome 19, rs429358 was the lead variant at the *APOE* locus ($\beta = 0.013$, $p = 3.9 \times 10^{-45}$), MAF = 15.6%. We tested whether the presence of *APOE-ε4* was enriched in certain primary causes of death. Among the top four causes of death, each representing over 5% of total deaths (Fig. 1B), only those due to "Diseases of the circulatory system" (Chi-square $p = 1.6 \times 10^{-16}$) and "Diseases of the nervous system" ($p = 1.1 \times 10^{-71}$) showed a significant enrichment in the proportion of ε4 carriers compared to the prevalence of ε4 carriers among all subjects (Fig. 1C). On chromosome 6 locus overlapping *ZSCAN23*, the top genome-wide significant variant was rs6902687, located 2.2 kb upstream of the transcription start site (TSS) (rs6902687_C: $\beta = 0.004$, $p = 2.7 \times 10^{-8}$, MAF = 36.6%). This variant is in almost perfect linkage disequilibrium ($R^2 > 0.99$) with three other significant variants in this region, including rs13215804_G (located 4.2 kb upstream of the TSS), rs111859903_G (located in an intron) and rs13190937_A (situated in the 5' untranslated region) (Fig. 1D).

To explore a potential regulatory function for variants at the *ZSCAN23* locus, we investigated whether the lead SNPs were expression quantitative trait loci (eQTLs) in the Genotype-Tissue Expression Project (GTEx) v8 database. rs13190937 was significantly associated with increased *ZSCAN23* expression in pancreatic tissue and the GWAS on Martingale residuals signal colocalized with the *ZSCAN23* expression quantitative trait loci (eQTL) (posterior probability of colocalization (PP4) = 0.934; Fig. 1E). Phenome-wide association study analysis (PheWAS) using PheWeb[14] based on UKB Neale v1 dataset shows that the main associations of rs13190937 are with celiac disease and intestinal malabsorption ($p = 1.8 \times 10^{-57}$, OR = 1.003) (Supplementary Fig. 1).

In sex-stratified GWAS (180,970 males and 212,863 females), the *APOE* locus was again linked to reduced lifespan in both males and females (Supplementary Table 1 and Supplementary Fig. 2A, B). In males, a significant association with reduced lifespan was observed for rs577106756_A located in intron of *PRKD3* on chromosome 2 ($\beta = 0.09$, $p = 3.2 \times 10^{-8}$, MAF = 0.1%). PheWAS analysis, based on the UKB Neale v1 dataset, revealed that rs577106756_A was associated with ICD10 code C10.9, Malignant neoplasm of oropharynx, unspecified, as the primary cause of death as the primary cause of death ($p = 4.5 \times 10^{-8}$, OR = 1.04), and self-reported "Stomach Cancer" ($p = 6.1 \times 10^{-7}$, OR = 1.003) (Supplementary Fig. 2C). Additionally, a borderline significant association with reduced lifespan was observed at rs35705950_T, located between *MUC5AC* and *MUC5B* on chromosome 11 ($\beta = 0.01$, $p = 6.6 \times 10^{-8}$, MAF = 11.2%) (Supplementary Fig. 2D). This variant was notably linked to increased *MUC5B* expression in lung tissue with the GWAS on Martingale residuals signal colocalizing with a *MUC5B* eQTL (PP4 = 0.99; Supplementary Fig. 2E). PheWAS analysis showed associations with a diagnosis of pulmonary fibrosis ($p = 4.4 \times 10^{-13}$), OR = 1.002), "Other interstitial pulmonary diseases with fibrosis" as the primary cause of death ($p = 1.7 \times 10^{-5}$, OR = 1.001), and paternal history of lung cancer ($p = 2.1 \times 10^{-4}$, OR = 1.004), but no association with maternal history of lung cancer ($p = 0.07$) (Supplementary Fig. 2F). In female, a significant association with reduced lifespan was observed for rs547541271_T, located in the intron of *CELF2* on chromosome 10 ($\beta = 0.04$, $p = 3.1 \times 10^{-8}$, MAF = 0.3%). PheWAS analysis indicated rs547541271_T was associated with a diagnosis of "Myositis"

($p = 1.0 \times 10^{-6}$, OR = 1.002), and self-reported Polycystic Ovaries/Polycystic Ovarian Syndrome ($p = 2.5 \times 10^{-5}$, OR = 1.003) (Supplementary Fig. 2G).

Further validation of these significant variants was carried out using data from the FinnGen and LifeGen cohorts. Specifically, for common variants, we queried FinnGen (https://r11.finngen.fi/) and obtained summary statistics[15] from the LifeGen consortium via the GWAS Catalog (https://www.ebi.ac.uk/gwas/downloads/summary-statistics) to assess their association in independent datasets. In FinnGen, rs13190937 was not significantly associated with the "Death" phenotype ($p = 0.2$), while it was significantly associated with a decrease in "Parental age at death" in the UKB and LifeGen consortium ($p = 1.4 \times 10^{-4}$, $\beta = -0.015$). Similarly, rs35705950 showed a significant association with increased death in FinnGen ($p = 6.0 \times 10^{-3}$, $\beta = 0.034$) and with decrease in "Parental age at death" in the UKB and LifeGen consortium ($p = 6.6 \times 10^{-3}$, $\beta = -0.023$). However, neither rs577106756 nor rs547541271 showed a significant association with the "Death" phenotype in FinnGen ($p = 0.55$ and 0.93, respectively) (Supplementary Table 2).

## Gene-based rare variant association analyses in whole-exome data

Among 26,230,624 variants with MAF < 1%, a total of 1,830,070 variants (17,174 genes) were annotated as loss-of-function (LoF) or missense variants. We excluded 199 genes with fewer than 10 total variant carriers, resulting in 476,447 predicted LoF variants (15,908 genes with a median of 23 variants per gene), 751,523 missense variants predicted as damaging by AlphaMissense (15,212 genes with a median of 37 variants per gene), and 262,866 missense variants predicted as damaging by rare exome variant ensemble learner (REVEL) (9231 genes with a median of 15 variants per gene). Of variants classified by each, 23.4% of AlphaMissense and 66.8% of REVEL variants were also pathogenic by the other classifier. A list of SNPs list used for the gene-based analyses is provided in Supplementary Data 1.

We identified six genes whose burden of LoF variants is significantly associated with reduced lifespan: *TET2* ($p = 2.6 \times 10^{-34}$), *ATM* ($p = 6.4 \times 10^{-10}$), *BRCA2* ($p = 1.2 \times 10^{-33}$), *CKMT1B* ($p = 4.3 \times 10^{-7}$), *BRCA1* ($p = 5.6 \times 10^{-12}$) and *ASXL1* ($p = 1.3 \times 10^{-51}$) (Fig. 2A and Table 1). All of these but *CKMT1B* also showed gene-wide significance in a direction-agnostic (SKAT-O) approach (Supplementary Fig. 3A). Additionally, in eight genes, the burden of missense variants predicted as pathogenic by AlphaMissense was associated with reduced lifespan: *DNMT3A* ($p = 6.9 \times 10^{-12}$), *SF3B1* ($p = 1.9 \times 10^{-13}$), *TET2* ($p = 9.2 \times 10^{-8}$), *PTEN* ($p = 1.6 \times 10^{-8}$), *SOX21* ($p = 2.2 \times 10^{-8}$), *TP53* ($p = 8.6 \times 10^{-17}$), *SRSF2* ($p = 1.8 \times 10^{-94}$) and *RLIM* ($p = 6.0 \times 10^{-7}$) (Fig. 2B). Lastly, three genes showed gene-wide significance for burden of missense variants predicted by REVEL: *DNMT3A* ($p = 5.2 \times 10^{-11}$), *PTEN* ($p = 1.2 \times 10^{-7}$), and *TP53* ($p = 2.2 \times 10^{-9}$) (Supplementary Fig. 4 and Supplementary Table 3). SKAT-O identified additional associations with pathogenic missense variants predicted by AlphaMissense in *C1orf52* ($p = 7.2 \times 10^{-8}$) and *IDH2* ($p = 5.4 \times 10^{-42}$) (Supplementary Fig. 3B), and by REVEL in *NMNAT2* ($p = 6.7 \times 10^{-7}$) and *TERT* ($p = 3.3 \times 10^{-10}$) (Supplementary Fig. 3C and Supplementary Table 3).

In addition, we validated these findings within the UKB dataset using two approaches: an independent sample separate from our discovery data and a five-fold cross-validation (CV) within the discovery cohort. This independent validation included 73,281 subjects who were not categorized as having European ancestry based on genetic ethnic grouping. These participants were classified into five groups based on their self-reported ethnicity (Field: 21000): White (66.3%), Asian (14.4%), Black (9.9%), Other (5.7%), and Mixed (3.7%). Among the 21 novel genes identified in the discovery, four achieved significance under the Bonferroni correction threshold $1.1 \times 10^{-3}$ (0.05/42) in this validation cohort: *BRCA2* ($p = 1.1 \times 10^{-3}$, burden), *ASXL1* ($p = 1.2 \times 10^{-5}$, burden; $p = 6.7 \times 10^{-6}$, SKAT-O) with LoF variants

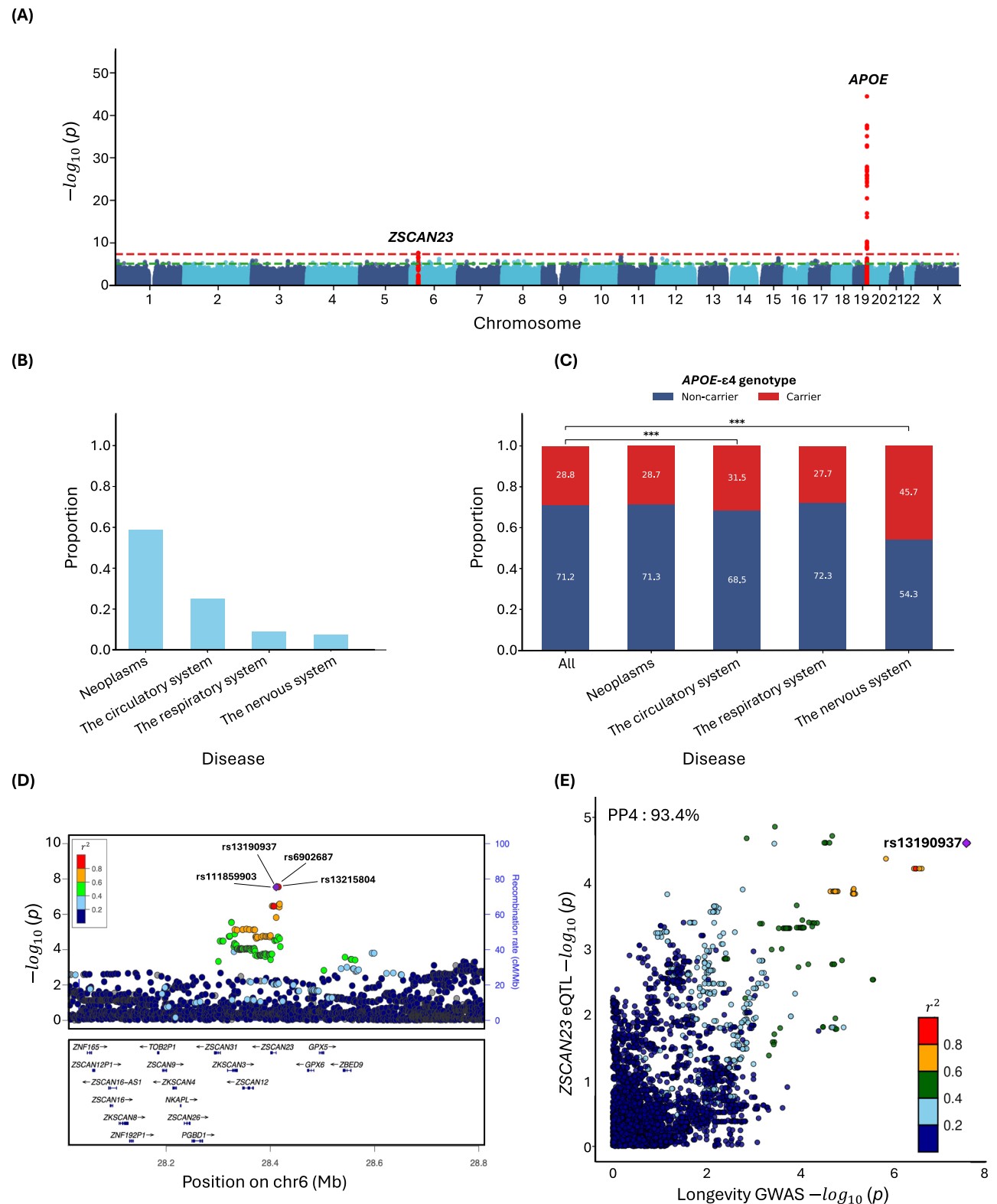

**Fig. 1 | Association of Common Variants with Lifespan. A** Manhattan plot. **B** The proportion of cause of death for the top 4 categories, each accounting for more than 5% of total deaths. **C** Association of causes of death with *APOE-ε4* genotype. **D** LocusZoom and **E** colocalization plots at the *ZSCAN23* locus, colocalized with *ZSCAN23* eQTL in pancreatic tissue in GTEx. PP4 posterior probability of colocalization.

and (*IDH2* $p = 2.0 \times 10^{-7}$, SKAT-O) and *SRSF2* ($p = 9.2 \times 10^{-9}$, burden; $p = 2.2 \times 10^{-10}$), SKAT-O) with pathogenic missense variants predicted by AlphaMissense (Supplementary Table 4). To further validate our findings, we performed five-fold CV within the discovery dataset of 393,833 individuals, dividing it into five folds. For each fold, 80% of the

data (315,066 individuals) was used for analysis. The results across folds were highly consistent. For example, *TET2*, BRCA2, *BRCA1*, *ASXL1* (LoF), SF3B1, *DNMT3A*, *IDH2*, *TP53*, *SRSF2* (AlphaMissense) and *DNMT3A* (REVEL) achieved gene-wide significance across all five folds. Except for *CKMT1B* (LoF), *C1orf52*, *TET2*, *RLIM* (AlphaMissense) and

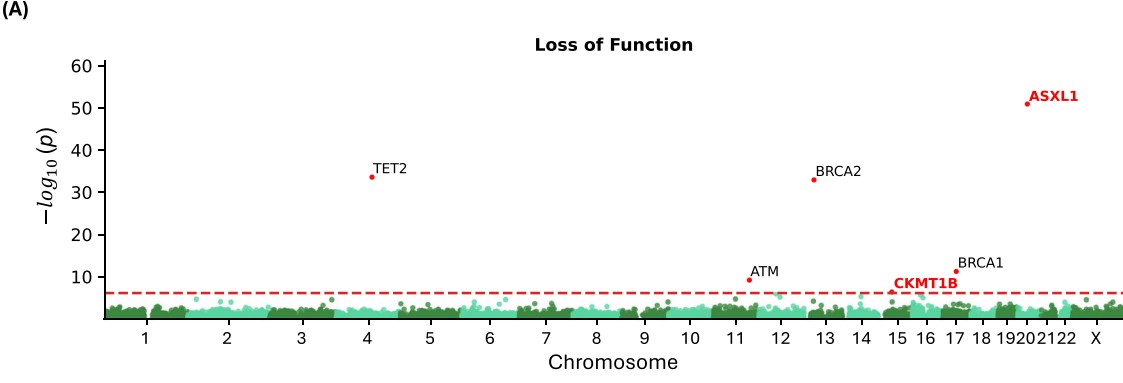

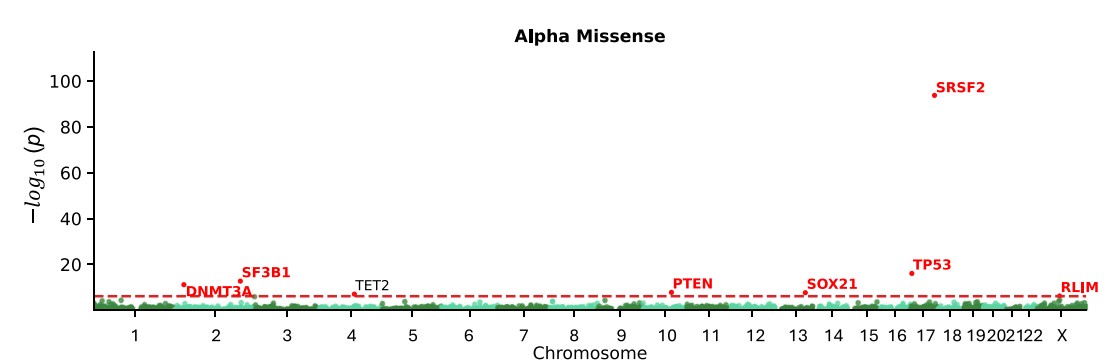

**Fig. 2 | Association of Rare Variant Burden with Lifespan.** Rare variant burden association with lifespan, considering loss-of-functions (**A**) and AlphaMissense pathogenic variants (**B**). Genes highlighted in red represent those not previously identified as significant in ref. 8. A gene-wide significance threshold of $p = 7.4 \times 10^{-7}$ was applied.

*NMNAT2* (REVEL), all other genes that showed significance in the main analysis were significant in at least 3 out of 5 folds. This consistency across folds confirms the robustness of the associations identified in our study. Fold-specific results for each gene and variant category are provided in Supplementary Data 2.

For sex-specific gene-based analysis, an additional four genes not identified in the whole-cohort analysis showed gene-wide significance in males by either burden or SKAT-O: *CDKN1A* and *PTPRK* (LoF); *COA7* and *TG* (AlphaMissense) (Supplementary Figs. 5A, 6A and Supplementary Table 5). In females, we identified five additional genes associated with reduced lifespan: *PORCN* (AlphaMissense); *UGT1A8*, *CBX3*, *IFITM10*, and *OLIG1* (REVEL) (Supplementary Figs. 5B, 6B and Supplementary Table 6).

**Gene-burden survival analysis**

For the 14 gene-wide significant genes in the burden analyses, we assessed the association of variant carrier status with lifespan using Cox proportional hazards regression. Carriers of LoF variants in six genes were associated with decreased survival compared to non-carriers: *CKMT1B* (HR = 3.9, $p = 2.6 \times 10^{-6}$), *ASXL1* (HR = 2.5, $p = 6.2 \times 10^{-33}$) (Fig. 3A), *TET2* (HR = 2.3, $p = 1.9 \times 10^{-22}$), *ATM* (HR = 1.7, $p = 3.0 \times 10^{-10}$), *BRCA2* (HR = 2.4, $p = 9.7 \times 10^{-39}$), and *BRCA1* (HR = 2.2, $p = 1.3 \times 10^{-12}$) (Supplementary Fig. 7A). Similarly, carriers of AlphaMissense-predicted pathogenic variants exhibited significantly earlier mortality compared to non-carriers on the following genes: *DNMT3A* (HR = 1.5, $p = 1.4 \times 10^{-9}$), *SF3B1* (HR = 2.3, $p = 4.4 \times 10^{-10}$), *PTEN* (HR = 4.0, $p = 1.1 \times 10^{-9}$), *SOX21* (HR = 1.9, $p = 3.3 \times 10^{-8}$), *TP53* (HR = 3.9, $p = 1.9 \times 10^{-14}$), *SRSF2* (HR = 5.8, $p = 3.3 \times 10^{-61}$), *RLIM* (HR = 3.1, $p = 2.9 \times 10^{-4}$) (Fig. 3B) and *TET2* (HR = 1.5, $p = 8.4 \times 10^{-7}$) (Supplementary Fig. 7B). Carriers of pathogenic variants predicted by REVEL showed similar trends: *DNMT3A* (HR = 1.6, $p = 2.4 \times 10^{-8}$), *PTEN* (HR =

4.8, $p = 1.0 \times 10^{-9}$), and *TP53* (HR = 2.5, $p = 1.5 \times 10^{-8}$) (Supplementary Fig. 7C).

To explore the contribution of individual rare variants to mortality in each gene-wide significant gene in the burden and SKAT-O tests, we conducted Cox proportional hazards regression for each variant with a minor allele count (MAC) of three or more (Table 2). In total, 587 variants including LoF, AlphaMissense, and REVEL variants were examined. After applying a Bonferroni correction for multiple testing, setting the significance threshold at $8.3 \times 10^{-5}$ (0.05/599), we identified significant associations with reduced lifespan for four LoF variants: rs370735654 in *TET2* (MAC = 17, HR = 7.9, $p = 6.1 \times 10^{-10}$), rs587779834 in *ATM* (MAC = 113, HR = 2.5, $p = 3.1 \times 10^{-5}$), rs80359705 in *BRCA2* (MAC = 13, HR = 11.4, $p = 2.5 \times 10^{-9}$), and rs750318549 in *ASXL1* (MAC = 201, HR = 2.8, $p = 3.5 \times 10^{-19}$). Additionally, significant associations with AlphaMissense variants were noted in seven genes, impacting lifespan: rs769009649 in *C1orf52* (MAC = 62, HR = 3.3, $p = 3.4 \times 10^{-7}$), rs147001633 in *DNMT3A* (MAC = 269, HR = 1.8, $p = 8.8 \times 10^{-6}$), rs377023736 in *SF3B1* (MAC = 12, HR = 7.3, $p = 2.5 \times 10^{-9}$), rs121913502 in *IDH2* (MAC = 45, HR = 6.8, $p = 9.2 \times 10^{-25}$), rs11540652 in TP53 (MAC = 5, HR = 11.5, $p = 2.3 \times 10^{-5}$), rs751713049 in *SRSF2* (MAC = 51, HR = 6.7, $p = 9.3 \times 10^{-31}$) and rs75871009 in *RLIM* (MAC = 6, HR = 6.2, $p = 6.0 \times 10^{-7}$). For missense variants predicted by REVEL, rs201746612 in *NMNAT2* (MAC = 5, HR = 11.0, $p = 1.7 \times 10^{-6}$), rs1043358053 in *TERT* (MAC = 5, HR = 16.8, $p = 1.7 \times 10^{-8}$), and rs11540652 in *TP53* (MAC = 5, HR = 11.5, $p = 2.3 \times 10^{-5}$) were significantly linked to reduced lifespan (Supplementary Table 7).

**Phenome-wide association studies**

For the nine novel genes identified in the burden test (*CKMT1B*, *ASXL1*, *DNMT3A*, *SF3B1*, *PTEN*, *SOX21*, *TP53*, *SRSF2* and *RLIM*), we examined the burden of LoF or pathogenic missense variants through PheWASs

**Table 1 | Significant genes for rare variants association with burden and SKAT-O tests ($p < 7.4 \times 10^{-7}$)**

| Variant class | Chr | Gene | # of variants | # of carriers | Burden p value | SKAT-O p value |
|---|---|---|---|---|---|---|
| LoF | 4 | *TET2* | 243 | 563 | $2.6 \times 10^{-34}$ | $1.2 \times 10^{-60}$ |
| | 11 | *ATM* | 247 | 1170 | $6.4 \times 10^{-10}$ | $5.0 \times 10^{-11}$ |
| | 13 | *BRCA2* | 245 | 1271 | $1.2 \times 10^{-33}$ | $3.5 \times 10^{-41}$ |
| | 15 | **CKMT1B** | 15 | 40 | $4.3 \times 10^{-7}$ | $1.5 \times 10^{-6}$ |
| | 17 | *BRCA1* | 120 | 456 | $5.6 \times 10^{-12}$ | $1.6 \times 10^{-11}$ |
| | 20 | **ASXL1** | 72 | 533 | $1.3 \times 10^{-51}$ | $6.8 \times 10^{-54}$ |
| AlphaMissense | 1 | **C1orf52** | 23 | 175 | $2.5 \times 10^{-5}$ | $7.2 \times 10^{-8}$ |
| | 2 | **DNMT3A** | 167 | 1229 | $6.9 \times 10^{-12}$ | $6.7 \times 10^{-13}$ |
| | 2 | **SF3B1** | 64 | 195 | $1.9 \times 10^{-13}$ | $2.6 \times 10^{-18}$ |
| | 4 | *TET2* | 159 | 826 | $9.2 \times 10^{-8}$ | $1.4 \times 10^{-7}$ |
| | 10 | **PTEN** | 50 | 71 | $1.6 \times 10^{-8}$ | $5.9 \times 10^{-11}$ |
| | 13 | **SOX21** | 52 | 463 | $2.2 \times 10^{-8}$ | $3.2 \times 10^{-8}$ |
| | 15 | **IDH2** | 89 | 349 | $1.4 \times 10^{-4}$ | $5.4 \times 10^{-42}$ |
| | 17 | **TP53** | 35 | 90 | $8.6 \times 10^{-17}$ | $6.2 \times 10^{-17}$ |
| | 17 | **SRSF2** | 14 | 141 | $1.8 \times 10^{-94}$ | $1.9 \times 10^{-114}$ |
| | X | **RLIM** | 25 | 51 | $6.0 \times 10^{-7}$ | $2.9 \times 10^{-9}$ |

Genes names in bold font represent those not previously identified as significant in ref. 8.
*LoF* loss of function, *Chr* chromosome.

across 1670 UKB phenotypes including disease occurrences derived from electronic health record, self-reported family history, and physical measures (Supplementary Fig. 8). The burden of LoF variants in *ASXL1* and AlphaMissense variants in *DNMT3A, SF3B1, PTEN, TP53* and *SRSF2* were strongly linked to an increased risk of leukemia: acute myeloid leukemia (*ASXL1*: Odds Ratio (OR) = 1.05; $p = 8.6 \times 10^{-170}$; *DNMT3A*: OR = 1.03, $p = 2.1 \times 10^{-150}$; *SRSF2*: OR = 1.3, $p = 1.2 \times 10^{-195}$; *TP53*: OR = 1.05, $p = 4.7 \times 10^{-35}$), monocytic leukemia (*DNMT3A*: OR = 1.01, $p = 2.5 \times 10^{-9}$), chronic lymphoid leukemia (*SF3B1*: OR = 1.07, $p = 4.1 \times 10^{-68}$) and acute lymphoid leukemia (*PTEN*: OR = 1.01, $p = 2.1 \times 10^{-14}$). Additionally, the burden of LoF in *CKMT1B* was associated with hypopharynx cancer (OR = 1.03, $p = 3.9 \times 10^{-26}$), vertiginous syndromes (OR = 1.03, $p = 3.0 \times 10^{-17}$) and salivary glands cancer (OR = 1.03; $p = 3.2 \times 10^{-12}$). *SOX21* burden was associated with increased acne (OR = 1.01, $p = 6.9 \times 10^{-7}$) and spinocerebellar disease (OR = 1.01, $p = 2.3 \times 10^{-6}$). Lastly, the burden of AlphaMissense variants in *RLIM* were associated with chromosomal anomalies and genetic disorders (OR = 1.02, $p = 4.2 \times 10^{-16}$), other and unspecified congenital anomalies (OR = 1.02, $p = 8.2 \times 10^{-11}$), malignant neoplasm of small intestine, including duodenum (OR = 1.02, $p = 3.5 \times 10^{-6}$) and cancer of oropharynx (OR = 1.02, $p = 1.3 \times 10^{-5}$).

### Somatic mutation and clonal hematopoiesis of indeterminate potential

We computed the variant allelic fraction (VAF) per carrier for each variant included in the analysis. Generally, germline variants have a mean VAF close to 50%, while somatic variants' mean VAF will be lower[16]. Thus, when an association is linked to clonal hematopoiesis of indeterminate potential (CHIP), we expect the distribution of VAF to be left-shifted compared to a normal distribution centered at VAF = 50%. Considering LoF variants, *TET2* (mean VAF across variants [95% bootstrap confidence interval for the mean VAF] = 0.33 [0.31,0.34]) and *ASXL1* (mean VAF = 0.32 [0.31,0.33]) burden test associations are supported by variants with a left-shifted VAF distribution (Supplementary Table 6 and Supplementary Fig. 9A). Similarly, considering pathogenic AlphaMissense variants, in *DNMT3A* (mean VAF = 0.24 [0.23–0.24]), *TET2* (mean VAF = 0.36 [0.34,0.38]), *TP53* (mean VAF = 0.28 [0.24,0.34]), *SRSF2* (mean VAF = 0.30 [0.28,0.31]) and *SF3B1* (mean

VAF = 0.31 [0.26,0.37]) are also left-shifted and the observed associations may be linked to CHIP (Supplementary Table 8 and Supplementary Fig. 9B).

## Discussion

In this study, we report several known and novel findings related to genetic risks associated with lifespan analyzing 393,833 European participants from the UKB. In the common variant GWAS, five independent loci associated with increased mortality risk were identified. In the gene-based analysis of rare non-synonymous variants, 16 genes were associated with lifespan via burden or SKAT-O tests.

Consistent with previous reports, rs429358, determining the *APOE-ε4* allele dosage, was associated with decreased lifespan across both sexes. *APOE-ε4* is well known for its associations with Alzheimer's Disease[17] and cardiovascular disease[18]. In our dataset, the proportion of ε4 carriers was significantly higher for deaths caused by "Disease of the circulatory system" and "Diseases of the nervous system" compared to the general prevalence of ε4 carriers, which could explain the effect of ε4 on lifespan. Examining the subcategories of these ICD-10 chapters, "Disease of the circulatory system" includes cardiovascular disease (I51.6), while "Diseases of the nervous system" covers Alzheimer's disease (G30). We also identified a genome-wide association at the *ZSCAN23* locus, which, while not previously reported in human lifespan studies, has been associated with rheumatoid arthritis[19,20], multiple sclerosis[19], and COVID-19[20] in other studies. Our colocalization analysis revealed that the lifespan-associated signal colocalizes with a *ZSCAN23* eQTL in pancreatic tissue with increased expression observed in minor allele carriers. Although the role of *ZSCAN23* remains unclear, recent studies have linked its expression to pancreatic tumors, supporting our colocalization findings[21]. Sex-specific GWAS in males identified a significant association in PRKD3, which PheWAS linked to neoplasms and stomach cancer. PRKD3 has been highlighted as a potential oncogene in various cancer types[22,23]. Additionally, a borderline significant association was found in males between *MUC5AC* and *MUC5B*, which highly colocalizes with a *MUC5B* eQTL in lung tissue, and several studies have linked this variant to pulmonary disease like idiopathic pulmonary fibrosis[24,25] and COVID-19[26,27]. In females, sex-stratified GWAS identified a variant in *CELF2* associated with reduced lifespan.

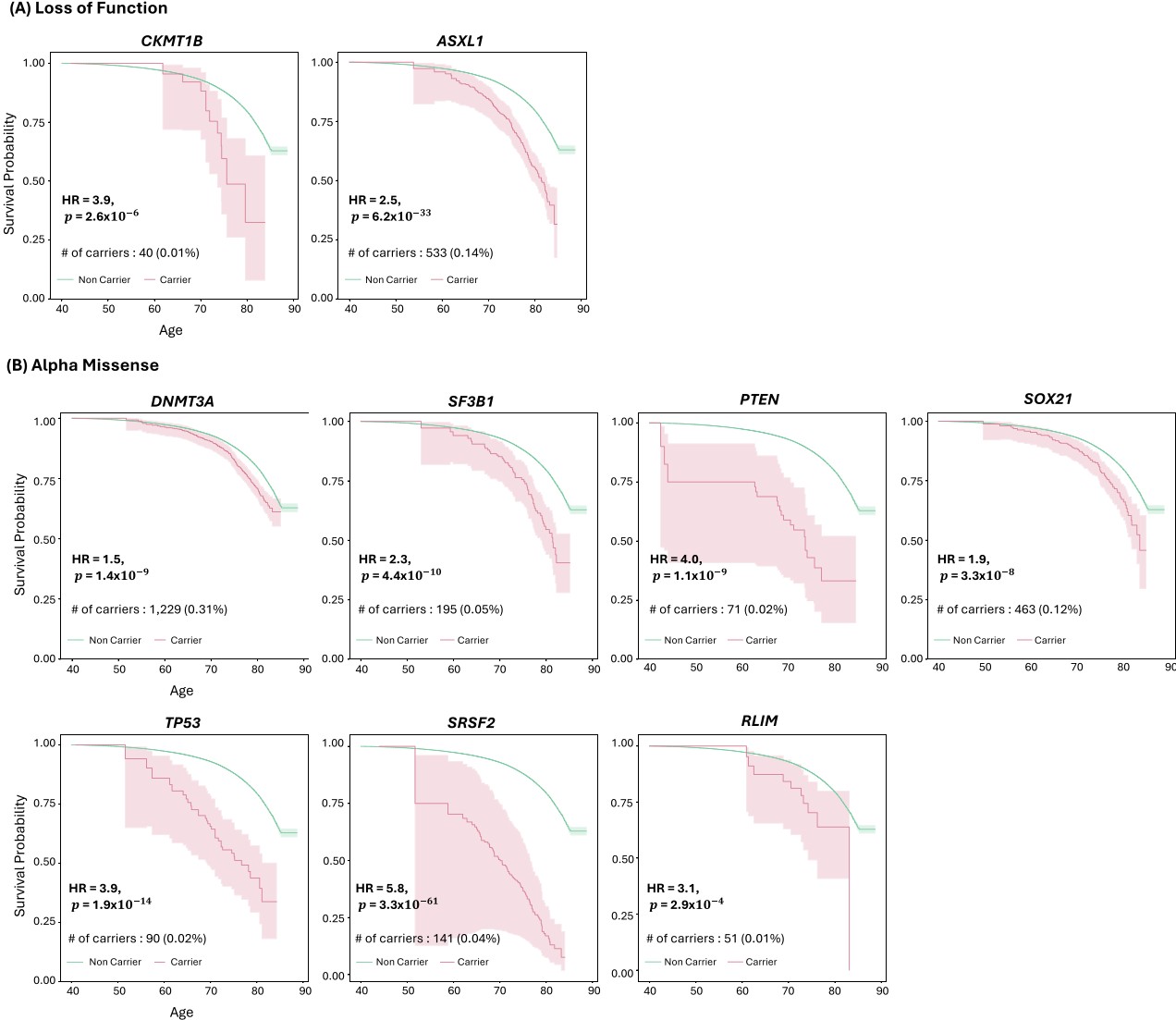

**Fig. 3 | Survival Analysis of Carriers vs Non-Carriers for Burden of Variants in Significant Genes.** Survival curves comparing carriers and non-carriers of variants on genes with a significant burden of loss-of-function (**A**) and AlphaMissense pathogenic (**B**) variants. For each gene, the survival curve includes Cox regression hazard ratio (HR), *p* value, the number of carriers, and their proportion within the total sample. A gene-wide significance threshold of $p = 7.4 \times 10^{-7}$ was applied.

*CELF2*, an RNA-binding protein, is a candidate gene for certain neurological disorders[28–30], and its activity has also been implicated in the development of ovarian and breast cancers[31,32]. Previously reported SNP associations with lifespan were concordant in our dataset but none of these passed the GWAS suggestive threshold ($p = 1.0 \times 10^{-5}$) except for those at the *APOE* locus (Supplementary Table 9). This phenomenon likely resulted from previous studies relying on proxy data such as parental age at death, which may capture a different set of genetic factors than direct proband mortality data.

In our gene-based rare variant analysis, 16 genes achieved gene-wide significance ($p < 7.4 \times 10^{-7}$) in either the burden or SKAT-O test. Four of these, *TET2, ATM, BRCA2,* and *BRCA1*, were reported in a previous rare-variant analysis of lifespan in UKB[8]. We identified 13 novel genes associated with lifespan—*CKMT1B, ASXL1, C1orf52, DNMT3A, SF3B1, PTEN, SOX21, IDH2, TP53, SRSF2, RLIM, NMNAT2* and *TERT*—when assessing variants causing genetic LoF or missense variants classified as pathogenic by REVEL or AlphaMissense. *ASXL1* was missed by a previous study considering protein truncating variants[8], as they excluded variants annotated as end-truncation, notably the main *ASXL1* variant driving CHIP (rs750318549 in Table 2). Of note, LoF and

missense variant analyses identified mostly separate genes with only one overlap (*TET2*). This supports the use of both categories in rare variant analyses and may indicate that missense variants as classified by AlphaMissense capture a wider range of variation missed when only assessing LoF variants, which are generally interpreted as resulting in haploinsufficiency. Importantly, missense variants may lead to increased or decreased protein function. In our analyses, *IDH2* was not gene-wide significant with the burden test ($p = 1.4 \times 10^{-4}$, Table 1) but was highly significant with SKAT-O ($p = 5.4 \times 10^{-42}$, Table 1). Since SKAT-O does not lose power when variants have differing directions of effect, this suggests that different mutations in *IDH2* can lead to either increased or decreased lifespan. These results underline the gain in information achieved when studying rare missense variants as well as LoF using appropriate statistical techniques.

Strikingly, most of the genes we identified carrying lifespan-associated rare variants have been previously linked to cancer. *TET2, ASXL1, DNMT3A,* and *SF3B1* are all known to harbor causal leukemia variants[33–36], and somatic variants in *SRSF2* have been described in myelodysplastic syndrome[37]. *ATM, BRCA2,* and *BRCA1* mutations have been well characterized in breast, ovarian, and other cancers[38–40]. *RLIM*

**Table 2 | Lead variant association per gene among significant genes in the burden and SKAT-O tests**

| Variant class | Chr | Gene | Variant | MA | MAC | AM | HR | *p* value | Reported |
|---|---|---|---|---|---|---|---|---|---|
| LoF | 4 | *TET2* | rs370735654 | T | 17 | – | 7.9 | $6.1 \times 10^{-10}$ | – |
| | 11 | *ATM* | rs587779834 | A | 113 | – | 2.5 | $3.1 \times 10^{-5}$ | Prostate Cancer[78] |
| | 13 | *BRCA2* | rs80359705 | A | 13 | – | 11.4 | $2.5 \times 10^{-9}$ | Breast Cancer[79] |
| | 15 | *CKMT1B* | rs1355844751 | T | 8 | – | 4.7 | $7.0 \times 10^{-3}$ | – |
| | 17 | *BRCA1* | rs80356991 | A | 11 | – | 3.7 | $3.4 \times 10^{-3}$ | Breast Cancer[80] |
| | 20 | *ASXL1* | rs750318549 | AG | 201 | – | 2.8 | $3.5 \times 10^{-19}$ | Leukemia[81] |
| AlphaMissense | 1 | *C1orf52* | rs769009649 | A | 62 | 0.876 | 3.3 | $3.4 \times 10^{-7}$ | – |
| | 2 | *DNMT3A* | rs147001633 | T | 269 | 0.995 | 1.8 | $8.8 \times 10^{-6}$ | Leukemia[82] |
| | 2 | *SF3B1* | rs377023736 | A | 12 | 0.999 | 7.3 | $2.5 \times 10^{-9}$ | – |
| | 4 | TET2 | rs76428136 | G | 5 | 0.913 | 9.4 | $1.0 \times 10^{-4}$ | – |
| | 10 | *PTEN* | rs587782350 | T | 3 | 0.941 | 14.7 | $7.2 \times 10^{-3}$ | Gastric Cancer[83] |
| | 13 | *SOX21* | rs1172148601 | A | 67 | 0.856 | 2.4 | $1.0 \times 10^{-3}$ | – |
| | 15 | *IDH2* | rs121913502 | T | 45 | 0.987 | 6.8 | $9.2 \times 10^{-25}$ | Leukemia[84] |
| | 17 | *TP53* | rs11540652 | T | 5 | 0.996 | 11.5 | $2.3 \times 10^{-5}$ | Breast Cancer[85] |
| | 17 | *SRSF2* | rs751713049 | T | 51 | 0.982 | 6.7 | $9.3 \times 10^{-31}$ | – |
| | X | *RLIM* | rs75871009 | G | 6 | 0.861 | 6.2 | $6.0 \times 10^{-7}$ | – |

Only variants with at least 3 minor alleles are reported. A significance threshold of $p = 8.3 \times 10^{-5}$ was applied after a Bonferroni correction for multiple testing. The "Reported" column indicates published studies that associated the highlighted variants with specific diseases, curated from ClinVar (https://www.ncbi.nlm.nih.gov/clinvar/).
*Chr* chromosome, *MAC* minor allele count, *AM* AlphaMissense score, *HR* hazard ratio.

appears to be a regulator of estrogen-dependent transcription, an important pathway in breast cancer[41], and has been recently described as a potential tumor suppressor[42]. *PTEN* and *TP53* are well studied due to their critical role in genomic stability and are the two most mutated genes in human cancer[43]. *IDH2* is also frequently mutated in many kinds of cancer[44]. The antisense long noncoding RNA *SOX21-AS1*, but not *SOX21*, has been linked to oral, cervical, and breast cancer[45–47]. A recent study found potential for *CKMT1B* expression as a prognostic biomarker in glioma[48]. *NMNAT2* expression has been found to be upregulated in colorectal cancer[49]. Finally, variation in both the coding and promoter sequences of *TERT* has been associated with a variety of cancer types[50,51].

Our PheWAS results also suggest that most of these genes are associated with cancer, specifically blood-based tumors such as myeloid leukemia. Combined with the common *ZSCAN23* locus we identified, associated with pancreatic tumors, this points to cancer being the major genetic factor currently affecting lifespan in UKB. This is consistent with a previous study of health span that found cancer to be the first emerging disease in over half of disease cases in UKB[52]. These results likely reflect the characteristics of the cohort, comprised of predominantly middle-aged individuals, with age-at-death ranging from 40.9 to 85.2 years and last-known ages between 52.6 and 88.7 years.

For sex-specific rare variant analyses, we identified four novel genes (*CDKN1A*, *PTPRK*, *COA7*, *TG*) in males and five genes (*PORCN*, *UGT1A8*, *CBX3*, *IFITM10* and *OLIG1*) in females. Some of these genes have been found to be associated with sex-specific diseases. In one study, advanced prostate cancer patients had a higher frequency of a variant on the 3′UTR of *CDKN1A*[53] and the gene has received attention as a potential therapeutic target for prostate cancer[54]. *PORCN* is located on the X chromosome and mutations on it can cause Goltz-Gorlin Syndrome[55], but it has also been found to regulate a signaling pathway that controls cancer cell growth[56]. *UGT1A8* expression is altered in endometrial cancer[57] and amino acid substitutions in it may modulate estradiol metabolism leading to an increased risk of breast and endometrial cancer[58].

Since UKB collected DNA from peripheral blood mononuclear cell samples, we explored whether the variants were potentially of somatic origin, picked up by WES genotyping due to CHIP. The VAF distribution of variants included in our analysis emphasizes that several associations are likely linked to CHIP and notably include the well-established CHIP-related genes *TET2, ASLX1, DNMT3A, SF3B1, TP53* and *SRSF2*. While WES heterozygote genotypes for these variants will not include all variants with some degree of CHIP within these genes, it does capture CHIP-related somatic variants sufficiently to establish robust associations with lifespan. In UKB the mean duration between the primary visit (blood draw date) and death is currently 9.2 years (± 3.8) and suggests that WES screening for CHIP variants may be used as a precision health tool to contribute to earlier cancer detection by assessing individuals with higher susceptibility risks[59]. In addition to known cancer variants, such as breast cancer-related *BRCA1/BRCA2*, our study highlights novel associations that should be considered in cancer susceptibility screenings.

While our study lacks an independent external replication and only a small number of genes formally replicated in the independent test set within UKB, the stability of the burden test association to five-fold CV suggests that these results are not due to outliers and are robust within the UKB.

By combining large-scale GWAS with rare variant analysis, this study enhances our understanding of the genetic basis of human lifespan. Our results emphasize the importance of understanding the genetic factors driving the most prevalent causes of mortality on a population level, highlighting the potential for early genetic testing to identify germline and somatic variants that place some individuals at risk of early death. Understanding the biological pathways through which these genes influence cancer and aging, as well as the environmental factors interacting with these pathways, will be essential for developing therapeutic targets aimed at extending a healthy lifespan. Our study's implications thus extend beyond genetics, as they touch on the broader aspects of health care, public health policy, and preventive strategies against age-related diseases.

In conclusion, this study enhances our understanding of the genetic basis of human lifespan by combining large-scale GWAS with detailed rare variant analysis. The novel loci identified warrant further exploration to understand their biological roles and interactions with environmental factors, which will be crucial for unraveling the

complex nature of aging and developing strategies to mitigate its adverse effects.

## Methods

### Study participants

The UKB is a large population-based longitudinal cohort study with recruitment from 2006 to 2010 in the United Kingdom[60]. In total, 502,664 participants aged 40–69 years were recruited and underwent extensive phenotyping including health and demographic questionnaires, clinic measurements, and blood draw at one of 22 assessment centers, of whom 468,541 subjects have been genotyped by both SNP array and WES.

We restricted our analysis to 393,833 individuals who self-reported their ethnic background as "white British" and were categorized as European ancestry based on genetic ethnic grouping (Field: 21000). Among them, 35,551 subjects were reported deceased, and their ages at death were recorded from the UK Death Registry (Field: 40007). For the other 358,282 subjects without death records, we assumed they were still alive by the latest censoring date (November 30, 2022). We determined their last known ages by subtracting their year and month of birth (Field: 33) from the censoring date.

All participants provided written informed consent, as outlined in the UK Biobank ethics framework (https://www.ukbiobank.ac.uk/learn-more-about-uk-biobank/governance/ethics-advisory-committee).

### SNP array genotyping and QC

A total of 488,000 UKB participants were genotyped using one of two closely related Affymetrix microarrays (UKB Axiom Array or UK BiLEVE Axiom Array) for approximately 850,000 variants. The genotyped dataset was phased using SHAPEIT3 and imputed with IMPUTE4, leveraging reference panels from UK10K, 1000 Genomes Project phase 3, and Haplotype Reference Consortium reference panels, resulting in approximately 97 million variants[61]. Additionally, we removed SNPs with imputation quality score <0.3, genotype missing rate >0.05, minor allele frequency (MAF) < 0.1%, and Hardy-Weinberg equilibrium $p < 1.0 \times 10^{-6}$.

### Genome-wide association studies

We performed linear regression analyses using BOLT-LMM (v.2.3.4)[62], which employs a linear mixed-effects model to test the association of common variants (MAF ≥ 0.1%) with lifespan for the entire cohort, as well as stratified by sex. For all three analyses, we used Martingale residuals calculated using the Cox proportional hazards model as the outcome variable. The procedure for calculating Martingale residuals was as follows. First, a Cox proportional hazards model[63] was fitted without genotype:

$$H(t|z) = H_0(t)e^{Z\beta'}$$

where $H_0(t)$ is the baseline hazard function at time point $t$ given the age at enrollment (Field: 54), last-known age and dead/alive status, $Z = [Z_1, \ldots, Z_k]$ is a covariate matrix, and $\beta = [\beta_1, \ldots, \beta_k]$ is a coefficient matrix for $Z$. Here, we included sex and the first 40 principal components (PC) and geographic covariates (Field: 20118) under the Geographical and Location category (100113) as covariates, but for sex-specific analyses, sex was excluded. Then, Martingale residuals were calculated as:

$$\widehat{M_i} = \delta_i - \widehat{H_0}(t)e^{Z\widehat{\beta}'}$$

where $\delta_i$ is the dead/alive status (0 = alive, 1 = dead) of the $i$ th subject and $\hat{\beta}$ is the estimated coefficient matrix. We adapted the *coxph* function from the *survival* (v.3.2.13)[64] in R (v.4.1.3) package to compute the Martingale residuals. Genome-wide significance threshold was set

at the standard GWAS level ($p = 5.0 \times 10^{-8}$). We used *LocusZoom*[65] to generate regional plots and Python v.3.7 to create Manhattan plots.

### Gene expression and colocalization analysis

To evaluate the effect of the significant loci identified in our GWAS, we examined expression quantitative trait loci (eQTLs) across 49 tissues having at least 73 samples from the Genotype-Tissue Expression Project (GTEx) version 8[66]. Bayesian colocalization analysis was employed using the *COLOC* package (v.5.2.3)[67] in R and the posterior probability of colocalization (PP4) was calculated between GWAS findings and eQTL associations within a 1 megabase (Mb) window. Additionally, colocalization was visualized using the *locuscompareR* package[68].

### Whole-exome sequencing and QC

Whole-exome sequencing (WES) data was available for 469,835 UKB participants. The dataset was generated by the Regeneron Genetics Center[69]. Details about the production and QC for the WES data was previously described[69]. We restricted the WES analysis to rare variants (MAF < 1%).

### Rare variant annotation

Rare variants in WES data were annotated using Variant Effect Predictor (v. 112) provided by Ensembl[70]. We defined LoF variants as those with predicted consequences: splice acceptor, splice donor, stop gained, frameshift, start loss, stop loss, transcript ablation, feature elongation, or feature truncation. Missense variants were annotated using AlphaMissense[71] and REVEL[72] plugins and included if they had an AlphaMissense score ≥ 0.7 or REVEL score ≥ 0.75. All annotation was conducted based on GRCh38 genome coordinates.

### Gene-based rare variant association studies

For testing groups of rare variants, genotype matrices were first transformed into a binary variable describing whether samples carry a variant of a given class as follows:

$$G_i = \begin{cases} 1, & \text{if } \sum_{j=1}^{k} g_{ij} > 0 \\ 0, & \text{if } \sum_{j=1}^{k} g_{ij} = 0 \end{cases}$$

Where $g_{ij}$ is the minor allele count observed for subject $i$ at variant $j$ in the gene and $k$ is the number of variants in the gene.

To account for relatedness and population structure, Martingale residuals were first adjusted using a linear mixed model approach implemented in fastGWA (--save-fastGWA-mlm-residual)[73]. The adjusted residuals were then used as the phenotype for the rare variant analyses, ensuring the robustness of the results in the presence of related individuals and population stratification.

We carried out two gene-based tests: the burden test and sequence kernel association test-optimized (SKAT-O)[74]. The burden test is a mean-based test that assumes the same direction of effects for all variants within a gene. On the other hand, SKAT-O employs a weighted average of the burden test and SKAT[75], the latter a variance-based test that does not lose power when variants have opposing directions of effect.

Association tests were performed for each gene and rare variant class separately, including LoF variants, missense variants with an AlphaMissense score ≥ 0.7, and missense variants with a REVEL score ≥ 0.75, using Martingale residuals as the phenotype as in the common variant analyses. We excluded genes with fewer than 10 variant carriers to ensure the reliability of our analyses. A gene-wide significance threshold was established at $p = 7.4 \times 10^{-7}$ based on the Bonferroni method accounting for the number of genes, variant classes, and statistical methods. Gene-based analyses were carried out using the *SKAT* package (v.2.2.5) in R.

To characterize the impacts of gene burden in significant genes, we compared lifespan survival depending on gene burden using Kaplan–Meier survival curves, and Cox proportional hazard regression analyses. Additionally, we performed Cox proportional hazards regression to assess the effect of each rare variant in a gene. The *survival* (v.3.2.13) package in R (v.4.1.3) was utilized for the survival analysis and the *lifelines* package (v.0.28.0) in Python v.3.7 was used for generating Kaplan–Meier survival curves.

## Phenome-wide association studies

For gene-wide significant genes, we conducted phenome-wide association studies (PheWAS) of variant carrier status across 1670 phenotypes in the UKB derived from binary, categorical, and continuous traits using the PHEnome Scan ANalysis Tool[76]. Phenotypes included the International Classification of Disease 10 (ICD-10) codes, family history (e.g., father's illness, father's age at death), blood count (e.g., white blood cell count), blood biochemistry (e.g., Glucose levels), infectious diseases (e.g., pp 52 antigen for Human Cytomegalovirus), physical measures (e.g., BMI), cognitive test (e.g., pairs matching) and brain measurements (e.g., subcortical volume of hippocampus). For ICD-10 codes, we excluded phenotypes from the following ICD-10 chapters: "Injuries, poisonings, and certain other consequences of external causes" (Chapter XIX), "External causes of morbidity and mortality" (Chapter XX), "Factors influencing health status and contacts with health services" (Chapter XXI), and "Codes for special purposes" (Chapter XXII). The ICD-10 codes were then converted into Phecodes (v.1.2)[77] which combine correlated ICD codes into a distinct code and improve alignment with diseases commonly used in clinical practice.

For binary traits, we removed phenotypes with fewer than 100 cases, and for continuous traits, those with fewer than 100 participants were excluded. Depending on the phenotype, we employed various regression models including binary logistic regression, ordinal logistic regression, multinomial logistic regression, and linear regression. All analyses included age and sex as covariates. Phenome-wide significance threshold was set at $p = 2.9 \times 10^{-5}$ based on the number of phenotypes.

## Variant allelic fraction

To investigate whether some gene-level associations are enriched for somatic variants, we computed the variant allele frequency (VAF) for each heterozygous sample, reporting the mean VAF and VAF distribution per gene per variant class. VAF is defined as the number of reads with an alternate allele divided by the read depth at a given variant position. We also calculated the confidence interval for the mean VAF per gene using 10,000 bootstrap samples to ensure robust statistical analysis.

## Reporting summary

Further information on research design is available in the Nature Portfolio Reporting Summary linked to this article.

# Data availability

GWAS summary statistics and rare variant results from SKAT-O and burden tests are available in the GWAS Catalog database under accession codes GCST90551884–GCST90551889. All phenotypic and genotypic data supporting the findings of this study are available from the UK Biobank (https://www.ukbiobank.ac.uk/enable-your-research/register). Access to these data is available from the authors with UKB permission.

# Code availability

The codes used for analyses in the present study are available at the following link: https://github.com/Junkkkk/Lifespan-studies. The code

for PheWAS analysis was utilized from https://github.com/MRCIEU/PHESANT.

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

## Acknowledgements

This research has been conducted using the UK Biobank Resource under application number 45420. We thank all the participants and researchers of UK Biobank for making these data open and accessible to the research community. This research was supported by the Dean's Postdoctoral Fellowship at the School of Medicine, Stanford University. Additionally, this research was partially supported by the Biostatistics Shared Resource (B-SR) of the NCI-sponsored Stanford Cancer Institute: P30CA124435 and by the following NIH funding source of Stanford's Center for Clinical and Translational Education and Research award, under the Biostatistics, Epidemiology and Research Design (BERD) Program: 1UM1TR004921-01.

## Author contributions

J.P. conducted all analyses, prepared all figures and drafted the manuscript. A.P.T. contributed to refining the manuscript. L.T. provided critical comment on the manuscript. M.D.G. and Y.L.G. planned, organized and supervised the entire study and revised the manuscript. All authors have approved the submitted version.

## Competing interests

The authors declare no competing interests.
