## [Transparent Peer Review file · Nature Communications]

Rare genetic associations with human lifespan in UK Biobank are enriched for oncogenic genes

Corresponding Author: Dr Junyoung Park

Version 0:

Reviewer comments:

Reviewer #1

(Remarks to the Author)

Park and colleagues report an interesting analysis of the UK Biobank cohort, studying the genetic associations with survival. The loci identified are enriched for cancer-associated oncogenes. This is in contrast to many previous publications of longevity (i.e., reaching very old age – not possible in the relatively young UK Biobank participants) or parental lifespan, which either just identify APOE (surely a contender for most pleiotropic locus after HLA) or predominantly cardiovascular/cardiometabolic pathways. The growing number of deaths in the UK Biobank cohort itself means the authors had the opportunity to study survival directly. Though as they note, the associations are predominantly driven by cancer, likely due to the age range of the cohort (i.e., although the max age is 89 in the follow-up, the median is 72). This does not undermine the associations identified, but leaves the door open for future studies of survival to identify further genes for longevity that are not “just” about avoiding death.

The methods are appropriate, to my knowledge (though I am not a particular expert on aggregate variant testing) and the conclusions justified. I really enjoyed reading the manuscript and commend the authors. I have a few minor comments and suggestions:

1. Please justify why you used a MAF $\geq 1\%$ cut-off for the common-variant analysis. Even variants with 0.1% MAF can be well-imputed in such large numbers.
2. Where a variant or gene appears to be “novel” for survival, such as rs13190937 in the 5' UTR of ZSCAN23, please report whether the variant (or a proxy) has been previously published. For example, rs13190937 appears in the GWAS catalog associated with Rheumatoid arthritis. Currently on line 344 you state “We also identified a GWS association at the ZSCAN23 locus, which had not been previously reported” which is therefore not true – it just hadn't been reported for survival before that I could see.
3. The results presented in Figure 3 are really striking, with some quite profound effect sizes (SRSF2 carriers have 25% survival probability by age 80!). But the text is too small to read. I suggest improving the layout and annotation of these figures, to really emphasize to readers what this is showing. Could you also include the frequency? E.g., what % of the population are carriers for the SRSF2 variants? I feel for me this is what is elevating the paper “beyond a GWAS.”

(Remarks on code availability)

The GitHub repository contains the main code to run the analysis. This is really positive.

Reviewer #2

(Remarks to the Author)

In the present study, the authors tested the association of common genetic variants and the burden of non-synonymous variants in a survival analysis in European ancestry participants of the UK Biobank. They identified a number of genes whose burden of loss-of-function variants or of pathogenic missense variants was significantly associated with reduced lifespan. Most of these genes have previously been linked to oncogenic-related pathways and/or are known to predispose to clonal

hematopoiesis. The methodology is sound and well-described and the results are of significance to the field and underline the importance of early genetic testing to identify germline and somatic variants with potential impact on cancer development and/or early death.

Comments are:

- In addition to known cancer variants such as BRCA1/BRCA2, the study describes novel associations that should be considered in cancer susceptibility screenings. Would the authors propose a specific panel for early cancer screening and what should that include? This includes some complex ethical questions such as genetic screening of healthy individuals, could the authors comment on that, how should genetic testing and communication with patients look like in the future?
- genes related to CHIP often show a very low VAF and generally, CHIP is diagnosed by the presence of somatic mutations at VAF of at least 2% in cancer-associated genes; what was the threshold used in the present study?
- mutations in TP53 are known to either be germline, somatically acquired in cancer or also related to CHIP; is it possible to distinguish between these detected missense variants?

(Remarks on code availability)

Reviewer #3

(Remarks to the Author)

Park *et al.* present a common variant genome-wide association study (GWAS) and rare variant gene-based study of participant survival in UK Biobank. They supplement these analyses with a genome-wide association scan (PheWAS) of significant genes and a test for clonal haematopoiesis of indeterminate potential (CHIP). The authors identify 3 common variant locus associations and 26 rare variant gene-based associations with participant survival (sex-stratified or combined), of which 2 loci and 22 genes have not been linked to (parental) lifespan before, respectively. Their findings represent a substantial increase in the number of genes linked to lifespan in UK Biobank. They conclude that many genes associated with participant survival are linked to CHIP and cancer.

The work by Park *et al.* closely parallels a 2022 Nature Aging study by Liu *et al.* (PMID: 37117740), who also performed gene-based rare variant burden analyses of participant survival in UK Biobank, and similarly performed sex-stratified, PheWAS, and CHIP analyses. While that study used exclusively protein-truncating variants in their burden tests, the current study extends this work by using loss-of-function and missense variants, as well as a direction-agnostic SKAT-O test, resulting in a wider search for lifespan genes with increased power to discover them.

However, I am concerned some of the findings by Park *et al.* may be false positives due to inadequate correction for participant relatedness, population stratification, and left-truncation. To illustrate the point, the authors discover a highly significant burden association for *ASXL* ($P = 3 \times 10^{-46}$) which was absent from the Liu *et al.* study, despite the lead variant (rs750318549) having a predicted frameshift consequence (i.e. protein-truncating) and could therefore have been detected in that study as well. This potential inflation of statistics is particularly problematic as the current study has not included any attempts at replication, making it hard to assess how reliable their findings and conclusions are. Finally, the authors have missed an opportunity to include non-European ancestries in their analyses.

Major concerns:

1. Participant relatedness was not accounted for in the association analyses, likely resulting in a considerable inflation of association statistics. The authors should explicitly take relatedness into account, either by excluding related individuals directly, or by fitting kinship as a random effect in their models. Regardless of their adjustment method, the authors should also estimate the degree of inflation in their common variant GWAS statistics using LD score regression or High-definition likelihood inference.
2. From my reading of the author-provided R code, the authors' Martingale residuals do not take into account left-truncation, and may therefore be subject to (larger) selection bias, which could skew downstream GWAS and gene burden statistics. Specifically, the authors report fitting the following model in R:

```
coxph(Surv(last_known_Age, Death) ~ Sex + PC1 + PC2 + PC3 + PC4 + PC5, data = df)
```

Where `last_known_Age` is the age of the participant at the censoring date (November 30, 2023). This model assumes participants were recruited at birth and followed up until the censoring date (i.e. right censored). However, UK Biobank participants were recruited only in middle age and could not be included in the study if they died before recruitment (i.e. selection). The authors should incorporate age at recruitment into the model, so the Martingale residuals more accurately reflect the survival of individuals after selection has taken place.

3. The authors only correct for five genetic principal components, raising a concern there is residual population stratification in the dataset which could inflate their association statistics. UK Biobank is highly geographically structured, and phenotypes commonly linked to lifespan (such as socioeconomic status and educational attainment) exhibit some of the strongest stratification across the country (PMID: 30659178, 31636407). At a minimum, the authors should adjust for geographic covariates (e.g. north and east coordinates) and the 40 genetic principal components provided by UK Biobank. Alternatively, they could fit the geographic variables and use a mixed model of participant kinship.

4. The authors have not used any independent internal or external datasets to replicate their findings, and failed to include any mention of non-European ancestries in their work. For common variant results, replication could be tested using publicly available GWAS statistics of participant survival in cohorts from various ancestries, including Biobank Japan (PMID: 36737517) and FinnGen (PMID: 36114182). Alternatively, some degree of replication could be achieved using parental lifespan GWAS, such as those from Ancestry.com (PMID: 31484785) or the LifeGen consortium (PMID: 29030599). While I appreciate datasets to replicate rare variant burden analyses are less readily available, an attempt could be made using cross-validation of the UK Biobank dataset.

Minor comments:

5. Line 51-52: The authors state that “According to previous studies, genetics accounts for as much as 40% of the heritability of longevity”; however, this is somewhat misleading as the most recent studies are the most robust and estimate the genetic component of human lifespan to be less than 10% (PMID: 30401766). Only outdated studies, which fail to take assortative mating into account, estimate it to be higher.

6. Line 55-58: Please be more specific about the phenotypes the loci are associated with, as there is a distinction between “longevity” (case/control - extreme survival), and “lifespan” (quantitative - survival) in the context of GWAS. I encourage the authors to be more clear with their language regarding these phenotypes throughout the manuscript to avoid confusion (PMID: 25814633). For example, the authors use the terms “longevity” and “Martingale residuals” interchangeably, but in genetic studies they represent distinct measurements.

7. Line 61-63: The authors state that “[proxy-based GWAS] may fail to comprehensively capture the genetic influences that directly impact an individual's lifespan” but the 2017 study by Liu *et al.* (PMID 28092683) which they cite does not necessarily support this argument. Liu *et al.* show that proxy-based GWAS effect estimates are directionally consistent with GWAS estimates of probands (albeit they need larger effective samples to achieve the same power). I acknowledge proxy-based GWAS may capture different genetic influences if the proxy samples are different (e.g. causes of death in the parental generation could be different from the current generation), but the authors should support this argument using more relevant citations.

8. The code provided by the authors is quite sparse and is missing the details of their phenotype quality control, the models used to fit Cox proportional hazard models of selected rare variants; and the entirety of the PheWAS. More detailed code would benefit the reproducibility of their study and the detail I can provide in my review.

9. At the time of review (14 September 2024), PheWeb results for rs13190937 also include “Disorders of iron metabolism” ($P = 2.9 \times 10^{-60}$), which is worth mentioning given the suspected link between iron metabolism and morbidity/mortality (PMID: 30651232, 32678081). I recommend including the PheWeb lookup date in the main text for transparency.

(Remarks on code availability)

The code provided by the authors is quite sparse and is missing the details of their phenotype quality control, the models used to fit Cox proportional hazard models of selected rare variants; and the entirety of the PheWAS. More detailed code would benefit the reproducibility of their study and the detail I can provide in my review.

Version 1:

Reviewer comments:

Reviewer #1

(Remarks to the Author)

Thanks to the authors for responding to my comments. I am happy to recommend the paper for publication. I look forward to seeing it in print!

(Remarks on code availability)

I have not been through every line but the repository is organised and I could find the information I was looking for. The authors should be commended for this!

Reviewer #2

(Remarks to the Author)

All my points were addressed, no further comments.

(Remarks on code availability)

Reviewer #3

(Remarks to the Author)

I appreciate the authors' explanation as to why the Liu *et al.* study did not identify *ASXL1*, and commend their efforts to update their analyses and improve the reliability of their study. Also, the expansion of the code provided by the authors has allowed me to more closely examine the methods, which has been a great help.

My only remaining concerns are regarding the rare variant analysis and its replication, and the exclusion of the X chromosome in the common variant GWAS. Otherwise, most of my outstanding comments focus on improving the clarity and interpretation of the findings.

Major concerns:

1) The use of BOLT-LMM has assuaged my concerns regarding statistical inflation due to participant relatedness in the common variant GWAS, but the authors make no mention of accounting for relatedness in the rare variant aggregation tests in the manuscript text and I am unable to find any such relatedness correction in their code. Previous studies which performed rare variant aggregation tests in UK Biobank invariably fit a genetic relatedness matrix (PMID: 36778668, 39210047) or remove related individuals (PMID: 31866045), so I would expect the authors to take family structure into account or justify why they chose not to account for it. I accept the genes highlighted by the authors plausibly affect lifespan through cancer incidence, but failure to account for relatedness could lead to overestimating the effects of these genes, and in turn, draw into question the authors' conclusion that preventative screening for such rare variants would have a material benefit. There are readily available tools like SAIGE-GENE+ and REGENIE which can perform rare variant aggregation tests in large cohorts while taking population structure and relatedness into account. Alternatively, the Martingale residuals phenotype itself could be adjusted for kinship using fastGWA's Gamma factor (--save-fastGWA-mlm-residual).

2) The authors use an independent UK Biobank sample to validate their rare variant results, but the genetic background of this validation sample consists of a mix of ancestries and, like the discovery sample, includes related individuals. Genetic associations are susceptible to population stratification, so ideally, each major genetic ancestry would be analysed separately and relatedness accounted for. I understand that rare variants require large samples to be testable, and splitting the validation sample by genetic ancestry reduces sample size, so I recommend at least one (but ideally both) of the following validation efforts:

2.1) Fit major genetic ancestry groups and genetic relatedness as covariates when testing the current multi-ancestry replication sample. Only genes significant in the UK Biobank discovery sample need to be tested for replication, so the multiple testing threshold does not have to be too stringent ($P < 0.05 / \text{number of significant genes tested for replication}$). In reporting their results, the authors should distinguish which genes could be tested for replication, which genes replicate at nominal $P < 0.05$ significance, and which genes replicate at multiple testing corrected significance.

2.2) Perform a leave-one-out cross-validation of the discovery dataset to check if the discovery effect estimates are stable. Specifically, the authors could leave out 20% of the British ancestry discovery sample and perform rare variant aggregation tests in the remaining 80% (accounting appropriately for relatedness as in my previous point). Repeating this analysis five times, where they exclude a different 20% of the sample in each iteration, should highlight which genes are stably associated with lifespan, and which genes are susceptible to population structure or bias in the discovery sample.

For the readers' clarity, I recommend the authors only mention replicated and/or cross-validated associations in the main text and include the full set of results in supplementary tables.

3) A closer look at the code suggests only autosomal chromosomes were tested for common variant GWAS associations and sex chromosomes were excluded, which is not explicitly stated in the main text. Excluding sex chromosomes is hard to justify when BOLT-LMM is able to test variants on the X chromosome (Y chromosome may be more complicated due to its non-recombining structure). For the rare variant analysis, the authors tested genes on the X chromosome, but I was unable to find the X chromosome-specific SKAT-O functions (SKAT_Null_Model_ChrX and SKAT_ChrX) in the author-provided code. Can the authors confirm the details of their rare variant X chromosome analysis?

Minor comments:

4) The revised text would benefit from a careful check to remove any typos and to confirm consistency with the updated results (e.g. line 92 says "two variants were GWS" but goes on to discuss three variants; line 102 has an unnecessary apostrophe before UK Biobank, "FinnGen" is misspelt as "FinnGenn", etc). Along the same lines, the heritability statement in the revised text differs from the statement mentioned in the rebuttal (Line 51-52). Please verify which is correct.

5) There is a typo in the Lifespan-studies/1.Martingale_residual/Define_last_known_age.py script, where the censoring year is mistakenly set to 2022 instead of the year 2023 reported in the manuscript text.

6) The author-provided code is missing sex-stratified versions of Martingale residual calculation, common variant GWAS, and rare variant burden analyses.

7) The code in the Lifespan-studies/3.Rare_variants_studies/Cox_regression_rare_variants.R script contains an adjustment for cluster(FID), implying correction for broad family relatedness when testing rare variant carrier status, but this covariate is not mentioned in the method section. Also, the Cox proportional hazards model in this script does not take into account left truncation.

8) Line 92-96: I believe the *ZSCAN23* SNPs rs111859903 and rs13190937 are almost in perfect linkage disequilibrium, so

mentioning each as a separate association in the main text seems unnecessary. For clarity, it would be helpful to annotate the *ZCAN23* locuszoom plot with all independent SNPs mentioned in the main text.

9) Line 122-128: It would be informative to mention the directionality in addition to significance when discussing replication of common variant results. For example, from my reading of the FinnGen data (<https://r11.finnngen.fi/variant/6:28443467-G-A>, accessed 6 November 2024), the direction of effect for the A allele of rs13190937 on 'Any death' is negative ($\beta = -0.011$, $p = 0.205$, i.e. increasing lifespan), while the effect on martingale residuals in the current study is positive ($\beta = +0.004$, $p = 2.9 \times 10^{-8}$, i.e. decreasing lifespan). In other words, the FinnGen association does not support the discovery. Note that it is also worth double-checking the numbers here as the trait name and P value I found differs from the one reported by the authors.

10) The replication analysis is made even harder to interpret when looking at the LifeGen + UKB associations, described as 'Parental Martingale residuals' in the main text but as 'Parental age at death' in Supplementary Table 2. The authors report a positive effect in the table for rs13190937 ($\beta = 0.015$, $P = 1.4 \times 10^{-4}$), suggesting a lifespan-increasing effect if the phenotype is age at death, which would contradict the discovery. More clarity in the text and table regarding directionality would greatly aid the reader.

11) In line with the previous point, reporting directionality for the PheWAS would be useful as well. For example, is the lifespan-reducing allele of rs13190937 associated with increased or decreased celiac disease? One would assume the former but it would be helpful to explicitly report the effect size and direction.

12) In line with Reviewer #1's comment on "novelty", figure legends would benefit from a more detailed description on what "novelty" means exactly. In addition, explicitly stating the sample sizes and significance thresholds in figure legends would make it easier for the reader to interpret the results.

13) In Supplementary Figure 1 and 2E, certain phenotypes have been underlined, but it is unclear from the legend why these phenotypes were highlighted.

14) There is no explanation in the Table 2 legend regarding the content of the "Reported" column: are these studies reporting associations with the gene or the variant? Are you allowing for proxy variants in high linkage disequilibrium? Moreover, it is unclear if the list of reported phenotypes is complete: the authors themselves show in their PheWAS analysis that many of the variants are associated with one or more phenotypes in Neale's analysis of UK Biobank (e.g. *CKMT1B* LoF and Hypopharynx cancer).

15) I suggest the authors edit the manuscript title to more accurately reflect the type of variants and the sample cohort used to identify oncogenic genes, as the current title appears to make the broad claim that all lifespan variants across all populations are enriched for oncogenic genes, while the study only identifies this link among rare variant associations in UK Biobank. One suggestion would be "Rare genetic associations with human lifespan in UK Biobank are enriched for oncogenic genes".

16) The discussion focuses largely on the findings but misses an opportunity to discuss the limitations of the study. For one, only a small number of variant and gene burden associations could be formally replicated, but this lack of replication is not acknowledged and the possible reasons for it are not discussed. Similarly, around line 392 the discussion would be enriched by including the authors' explanation of ASXL1 (as mentioned in their rebuttal), with the added takeaway that the selection of variants can strongly influence the findings, and is therefore an important limitation of the study.

17) Line 385-389: Which previous lifespan associations were checked? There is no mention of this analysis in the main text or any table to support the statement. Additionally, the authors state the phenomenon likely resulted from proxy vs. proband data, but an alternative explanation could be the differences in sample size. A quick calculation of the sample size needed to detect the previously reported effects could help determine to what degree sample size plays a role in the discrepancy.

18) Please provide a Supplementary Data file with the list of SNPs for each gene used to calculate burden and SKAT-O statistics. This will allow other researchers to more easily reproduce the authors' analysis.

(Remarks on code availability)

Version 2:

Reviewer comments:

Reviewer #3

(Remarks to the Author)

Thank you for taking the time to address my concerns. I can wholeheartedly recommend the updated manuscript for publication. To maximise the impact of this work, I highly recommend the authors release the full summary statistics for their rare variant analyses (in addition to the GWAS statistics already mentioned in their Data Availability statement). I also encourage the authors to double check these statistics are indeed publicly available, and to include DOIs or Study Accession Identifiers in the Data Availability Statement to make their statistics as easy to find as possible.

(Remarks on code availability)

Title: Genetic associations with human longevity are enriched for oncogenic genes

Responses to reviewers' comments

Please note that the revisions made have not influenced the content, conclusions, or framework of the paper. We have not listed all minor revisions made; however, these are highlighted in yellow in the revised manuscript.

Reviewer 1's comments

Park and colleagues report an interesting analysis of the UK Biobank cohort, studying the genetic associations with survival. The loci identified are enriched for cancer-associated oncogenes. This is in contrast to many previous publications of longevity (i.e., reaching very old age – not possible in the relatively young UK Biobank participants) or parental lifespan, which either just identifies APOE (surely a contender for most pleiotropic locus after HLA) or predominantly cardiovascular/cardiometabolic pathways. The growing number of deaths in the UK Biobank cohort itself means the authors had the opportunity to study survival directly. Though as they note, the associations are predominantly driven by cancer, likely due to the age range of the cohort (i.e., although the max age is 89 in the follow-up, the median is 72). This does not undermine the associations identified but leaves the door open for future studies of survival to identify further genes for longevity that are not “just” about avoiding death.

#1

Please justify why you used a MAF $\geq 1\%$ cut-off for the common-variant analysis. Even variants with 0.1% MAF can be well-imputed in such large numbers.

We originally chose a MAF $\geq 1\%$ cutoff as a commonly used threshold for “common” variants. Reviewer #1 is correct, however, regarding the imputation quality of most variants with MAF between 0.1% and 1%. Thus, we re-ran the analysis on imputed data to include variants with MAF between 0.1% and 1% and imputation quality ($R_{sq} > 0.3$). No additional genome-wide significant variants were identified in the non-sex-stratified analysis (see Manhattan plot below). Our results now also include variants with frequency between 0.1% and 1%.

The Manhattan plot below corresponds to variants with a frequency between 0.1% and 1% for the main analysis:

#2

Where a variant or gene appears to be “novel” for survival, such as rs13190937 in the 5' UTR of ZSCAN23, please report whether the variant (or a proxy) has been previously published. For example, rs13190937 appears in the GWAS catalog associated with Rheumatoid arthritis. Currently, on line 344 you state “We also identified a GWS association at the ZSCAN23 locus, which had not been previously reported” which is therefore not true – it just hadn't been reported for survival before that I could see.

The original statement was intended to indicate that the variant had not been previously reported in association with human lifespan, but we acknowledge that this was not clearly stated. As the reviewer correctly noted, rs13190937 has been reported in the GWAS catalog as associated with rheumatoid arthritis, multiple sclerosis, and COVID-19 in two studies. We have revised the sentence accordingly to reflect this clarification.

“We also identified a genome-wide association at the *ZSCAN23* locus, which, while not previously reported in human lifespan studies, has been associated with rheumatoid arthritis [1, 2], multiple sclerosis [1], and COVID-19 [2] in other studies.”

1. Wen, Y.-P. and Z.-G. Yu, *Identifying shared genetic loci and common risk genes of rheumatoid arthritis associated with three autoimmune diseases based on large-scale cross-trait genome-wide association studies*. *Frontiers in Immunology*, 2023. **14**.
2. Yao, M., et al., *Disentangling the common genetic architecture and causality of rheumatoid arthritis and systemic lupus erythematosus with COVID-19 outcomes: Genome-wide cross trait analysis and bidirectional Mendelian randomization study*. *Journal of Medical Virology*, 2023. **95**(2): p. e28570.

#3

The results presented in Figure 3 are really striking, with some quite profound effect sizes (SRSF2 carriers have 25% survival probability by age 80!). But the text is too small to read. I suggest improving the layout and annotation of these figures, to really emphasize to readers what this is showing. Could you also include the frequency? E.g., what % of the population are carriers for the SRSF2 variants? I feel for me this is what is elevating the paper “beyond a GWAS.”

We appreciate the positive feedback regarding Figure 3. We have revised the figure to improve text readability and layout, ensuring that the effect sizes are more clearly emphasized. Additionally, we have included the population frequencies for all variants to provide context for the survival probability data and further elevate the impact of the results (See below).

(A) Loss of Function

(B) Alpha Missense

Reviewer 2's comments

In the present study, the authors tested the association of common genetic variants and the burden of non-synonymous variants in survival analysis in European ancestry participants of the UK biobank. They identified a number of genes whose burden of loss-of-function variants or pathogenic missense variants was significantly associated with reduced lifespan. Most of these genes have previously been linked to oncogenic-related pathways and/or are known to predispose to clonal hematopoiesis. The methodology is sound and well-described and the results are of significance to the field and underline the importance of early genetic testing to identify germline and somatic variants with potential impact on cancer development and/or early death.

#1

In addition to known cancer variants such as BRCA1/BRCA2, the study describes novel associations that should be considered in cancer susceptibility screenings. Would the authors propose a specific panel for early cancer screening and what should that include? This includes some complex ethical questions such as genetic screening of healthy individuals, could the authors comment on that, how should genetic testing and communication with patients look like in the future?

We briefly touched upon these implications in our discussion:

“By combining large-scale GWAS with rare variant analysis, this study enhances our understanding of the genetic basis of human longevity. Our results emphasize the importance of understanding the genetic factors driving the most prevalent causes of mortality on a population level, highlighting the potential for early genetic testing to identify germline and somatic variants that place some individuals at risk of an early death. Understanding the biological pathways through which these genes influence cancer and aging, as well as the environmental factors interacting with these pathways, will be essential for developing therapeutic targets aimed at extending healthy lifespan. Our study's implications thus extend beyond genetics, as they touch on the broader aspects of health care, public health policy, and preventive strategies against age-related diseases.”

Given that early cancer identification leads to a better prognosis, we would recommend genetic screening as part of the standard of care and closer monitoring of individuals harboring high-risk variants. We agree that this may have implications beyond patient care and healthcare policy makers would have to regulate the use of these genetic data, notably to prevent detrimental usage by health insurance companies. Our study provides a list of significant genes and criteria on the variant types to monitor that should allow for a reduced cost, targeted whole-exome sequencing on this panel of genes. As noted, in our manuscript and in the following reviewer #2's comment, some of these variants are CHIP-related somatic variants and are picked up by the WES germline joint-calling pipeline due to a combination of higher VAF and high WES coverage. Thus, this genetic screening could be repeated every 5 years (after 50 years old) to re-assess an individual's CHIP status. If CHIP status or cancer susceptibility variants are identified, then an individual should be closely monitored for cancer to enable timely treatment.

#2

genes related to CHIP often show a very low VAF and generally, CHIP is diagnosed by the presence of somatic mutations at VAF of at least 2% in cancer-associated genes; what was the threshold used in the present study?

In our study, we solely consider genotype calls provided by the UK Biobank germline joint-calling pipeline. However, we do acknowledge that in this pipeline some heterozygous calls may be linked to the presence of somatic variants present in multiple cell clones expanded through CHIP. The VAF distribution per gene in carriers (defined by the WES genotype/ GT field) of these variants allows us to infer whether the observed gene-level association is supported by germline or most likely by underlying variants promoting CHIP. In the results section, we emphasize genes that have a left-shifted VAF distribution compared to a normal distribution centered at VAF = 50%. Evidence in the literature supports the implication of these genes in CHIP, thus we wrote that for these genes with left-shifted VAF, “the observed associations may be linked to CHIP”.

Reviewer #2 does bring up an important point, which is the presence of other individuals, who do have VAF > 2% for some of the considered variants but who have a homozygous genotype (GT = 0/0) in the WES joint call and thus are not being considered as carriers in our Burden test. We have not run an analysis considering VAF to re-genotype call individuals, as our design is a gene-wide burden/SKAT test on genotype calls provided by the UK Biobank. We also believe that individuals with a lower VAF (less likely to be called heterozygous) may be at an earlier CHIP stage at the time of their blood draw and may thus dilute the presented associations.

#3

mutations in TP53 are known to either be germline, somatically acquired in cancer or also related to CHIP; is it possible to distinguish between these detected missense variants?

Based on Figure S9 (B) showing VAF for carriers of *TP53* variants, most VAFs, except one carrier (See below table), are below 0.5, and most are below 0.4 which suggests that the variants considered for *TP53* (AlphaMissense > 0.7) are most likely CHIP-related.

Subject ID	SNP ID	VAF	DP
carrier 1	rs121913343	0.26	66
carrier 2	rs121913343	0.21	70
carrier 3	rs121913343	0.21	75
carrier 4	rs121913343	0.54	59
carrier 5	rs121913343	0.44	25
carrier 6	rs121913343	0.38	61
carrier 7	rs121913343	0.28	47
carrier 8	rs121913343	0.28	18
carrier 9	rs530941076	0.29	45
carrier 10	rs530941076	0.47	51
carrier 11	rs587782596	0.44	79
carrier 12	rs587782596	0.34	59
carrier 13	rs587780068	0.47	60

Also, gnomAD (<https://gnomad.broadinstitute.org/>) allelic imbalance for heterozygous individuals of these variants suggests that these variants picked up by WES/WGS are likely linked to CHIP (mostly picked up by WES in gnomAD).

[figure redacted]

Reviewer 3's comments

Park et al. present a common variant genome-wide association study (GWAS) and rare variant gene-based study of participant survival in UK Biobank. They supplement these analyses with a phenome-wide association scan (PheWAS) of significant genes and a test for clonal haematopoiesis of indeterminate potential (CHIP). The authors identify 3 common variant locus associations and 26 rare variant gene-based associations with participant survival (sex-stratified or combined), of which 2 loci and 22 genes have not been linked to (parental) lifespan before, respectively. Their findings represent a substantial increase in the number of genes linked to lifespan in UK Biobank. They conclude that many genes associated with participant survival are linked to CHIP and cancer.

The work by Park et al. closely parallels a 2022 Nature Aging study by Liu et al. (PMID: 37117740), who also performed gene-based rare variant burden analyses of participant survival in UK Biobank, and similarly performed sex-stratified, PheWAS, and CHIP analyses. While that study used exclusively protein-truncating variants in their burden tests, the current study extends this work by using loss-of-function and missense variants, as well as a direction-agnostic SKAT-O test, resulting in a wider search for lifespan genes with increased power to discover them.

However, I am concerned some of the findings by Park et al. may be false positives due to inadequate correction for participant relatedness, population stratification, and left-truncation. To illustrate the point, the authors discover a highly significant burden association for *ASXL* ($P = 3 \times 10^{-46}$) which was absent from the Liu et al. study, despite the lead variant (rs750318549) having a predicted frameshift consequence (i.e. protein-truncating) and could therefore have been detected in that study as well. This potential inflation of statistics is particularly problematic as the current study has not included any attempts at replication, making it hard to assess how reliable their findings and conclusions are. Finally, the authors have missed an opportunity to include non-European ancestries in their analyses.

Reviewer #3 raises several valid concerns that we addressed by responding to their comments below. We also noted the difference of significance for *ASXL1* between our study and the one of Liu et al.. The explanation is straightforward and presented in our manuscript. In brief, Liu et al. considered protein-truncating variants not flagged by LOFTEE and thus excluded several variants that we considered on *ASXL1*, including the most significant variant on *ASXL1* (reported in Table 2, flagged pLoF Low-confidence (END_TRUNC) https://gnomad.broadinstitute.org/variant/20-32434638-A-AG?dataset=gnomad_r4). GnomADv4.1 has an additional flag reporting discrepant frequencies between WES and WGS, and this is likely due to WES's higher read depth which picks up more CHIP genotype calls. As highlighted in our manuscript the association of *ASXL1* LoF variants with longevity in UKB WES is likely CHIP-related (*ASXL1* is a well-known CHIP gene, <https://www.nature.com/articles/s41467-021-22053-y>)

Liu et al., 2022 method section

“LOFTEE applies a range of filters on stop-gained, splice-site disrupting, and frameshift variants to exclude putative PTVs due to variant annotation and sequencing mapping errors that are unlikely to substantially disrupt gene function. For instance, stop-gained and frameshift variants that are within 50 kb of the end of the transcript will be flagged as ‘low confidence’. We extracted variants predicted as PTVs, flagged as ‘high confidence’ by LOFTEE”

While reviewer #3 makes some valid general points that one should be aware of to address statistical inflation, they probably had specific concerns regarding the high significance of *ASXL1* in the burden test on loss-of-function variants. However, among the other 4 reported genes by Liu et al. (*TET2*, *ATM*, *BRCA2*, *BRCA1*), three have significance in the same range in our study and their study, with the exception of *TET2*, which as for *ASXL1* is a known CHIP gene and for which we included additional variants compared to Liu et al.

Overall, our common variant analyses (both sexes, females-stratified, males-stratified) solely report *APOE* and two additional loci previously identified in disease/cancer-GWAS. The genomic inflation factors

(lambda median) are also within an expected range (1.018, 1.036, 1.05), and a close examination of QQ plots (See figures below) does not show evidence of statistical inflation.

However, to fully address Reviewer 3's concerns, we re-ran the analyses with left-truncation, 40 PCs, and a kinship matrix for additional statistical rigor.

Major concerns

#1

Participant relatedness was not accounted for in the association analyses, likely resulting in a considerable inflation of association statistics. The authors should explicitly take relatedness into account, either by excluding related individuals directly, or by fitting kinship as a random effect in their models. Regardless of their adjustment method, the authors should also estimate the degree of inflation in their common variant GWAS statistics using LD score regression or High-definition likelihood inference.

Although, as mentioned above, the previous methodology (model not accounting for relatedness) did not show overfitting in terms of genomic inflation factors (genomic inflation factor=1.01), we believe that a model incorporating kinship as a random effect would be more appropriate considering the characteristics of the UKB data. Therefore, for common variants, we re-ran the association analyses using BOLT-LMM (v.2.3.4).

#2

From my reading of the author-provided R code, the authors' Martingale residuals do not take into account left-truncation, and may therefore be subject to (larger) selection bias, which could skew downstream GWAS and gene burden statistics. Specifically, the authors report fitting the following model in R:

```
coxph(Surv(last_known_Age, Death) ~ Sex + PC1 + PC2 + PC3 + PC4 + PC5, data = df)
```

Where last_known_Age is the age of the participant at the censoring date (November 30, 2023). This model assumes participants were recruited at birth and followed up until the censoring date (i.e. right censored). However, UK Biobank participants were recruited only in middle age and could not be included in the study if they died before recruitment (i.e. selection). The authors should incorporate age at recruitment into the model, so the Martingale residuals more accurately reflect the survival of individuals after selection has taken place.

This is an excellent point. We re-calculated the martingale residuals, incorporating left truncation using the age at 1st visit (Field: 54) and adjusting for additional PCs (1-40) and geographic covariates (Field: 20118) under the Geographical and Location category (100113), as suggested below.

#3

The authors only correct for five genetic principal components, raising a concern there is residual population stratification in the dataset which could inflate their association statistics. UK Biobank is highly geographically structured, and phenotypes commonly linked to lifespan (such as socioeconomic status and educational attainment) exhibit some of the strongest stratification across the country (PMID: 30659178, 31636407). At a minimum, the authors should adjust for geographic covariates (e.g. north and east coordinates) and the 40 genetic principal components provided by UK Biobank. Alternatively, they could fit the geographic variables and use a mixed model of participant kinship.

We appreciate this valuable insight. We increased the number of principal components from 5 to 40. Unfortunately, the geographic coordinates suggested (Field: 22686-22689) are restricted data and inaccessible to us. However, we have included a closely related covariate accounting for city/area type (Field: 20118), which indicates the type of neighborhood participants live in, based on the characteristics of each residential area (e.g., urban/rural in England and Scotland) and population density. This variable was used, and categories were encoded with dummy variables.

As noted in our response to comment #2 above, we re-calculated the martingale residuals by considering left truncation and adding covariates and subsequently re-conducted all analyses (GWAS for common variants and gene-based tests, survival analysis for rare variants). We have updated the main text, tables, figures, and supplementary materials accordingly. The significant results did not differ from the original findings.

#4

The authors have not used any independent internal or external datasets to replicate their findings, and failed to include any mention of non-European ancestries in their work. For common variant results, replication could be tested using publicly available GWAS statistics of participant survival in cohorts from various ancestries, including Biobank Japan (PMID: 36737517) and FinnGen (PMID: 36114182). Alternatively, some degree of replication could be achieved using parental lifespan GWAS, such as those from [Ancestry.com](https://www.ancestry.com) (PMID: 31484785) or the LifeGen consortium (PMID: 29030599). While I appreciate datasets to replicate rare variant burden analyses are less readily available, an attempt could be made using cross-validation of the UK Biobank dataset.

(Response) For common variants, we queried FinnGenn (<https://r11.finnngen.fi/>) and downloaded summary statistics from the LifeGen consortium via GWAS catalog (<https://www.ebi.ac.uk/gwas/downloads/summary-statistics>) for the significant common variants. We note that rs13190937 is associated with the 'Death' phenotype in FinnGen ($p=0.28$) and with 'Parental Martingale residuals' in the UKB and LifeGen consortium ($p=1.4e-4$). Additionally, rs35705950 showed an association with death in FinnGen ($p=6.0e-3$) and with 'Parental Martingale residuals' in the UKB and LifeGen consortium ($p=6.6e-3$). Lastly, rs547541271 was not significantly associated with the 'Death' phenotype in FinnGen ($p=0.93$). We have added the above content to the main text and included the corresponding results in Supplementary Table 2.

For rare variants, as noted by reviewer #3, we are not aware of any large-scale, population-based WES datasets that are publicly available and could be used to independently validate our rare variant burden analyses. Therefore, we conducted validation using cross-validation within the UK Biobank dataset, employing an independent sample separate from our discovery data. A total of 75,473 subjects, who were not categorized as having European ancestry based on genetic ethnic grouping (Field: 21000), were included. In the validation dataset, two of the 16 novel genes identified in the discovery dataset achieved gene-wide significance with pathogenic missense variants predicted by AlphaMissense in IDH2 ($p=4.3e-7$, SKAT-O) and SRSF2 ($p=4.0e-10$, burden; $p=2.8e-11$, SKAT-O). *ASXL1* also approached gene-wide

significance in both the burden ($p=7.7e-6$), and SKAT-O ($p=5.5e-6$) analyses. We have added the above content to the main text and included the corresponding results in Supplementary Table 4.

Minor concerns

#5

Line 51-52: The authors state that “According to previous studies, genetics accounts for as much as 40% of the heritability of longevity”; however, this is somewhat misleading as the most recent studies are the most robust and estimate the genetic component of human lifespan to be less than 10% (PMID: 30401766). Only outdated studies, which fail to take assortative mating into account, estimate it to be higher.

We rephrased the heritability statement as

“According to previous studies, genetics accounts for less than 10% (PMID: 30401766) or up to 25% of the heritability of longevity (PMID: 28689042).”

#6

Line 55-58: Please be more specific about the phenotypes the loci are associated with, as there is a distinction between “longevity” (case/control - extreme survival), and “lifespan” (quantitative - survival) in the context of GWAS. I encourage the authors to be more clear with their language regarding these phenotypes throughout the manuscript to avoid confusion (PMID: 25814633). For example, the authors use the terms “longevity” and “Martingale residuals” interchangeably, but in genetic studies they represent distinct measurements.

We revised our manuscript to use ‘lifespan’ or ‘GWAS on Martingale residuals’ in place of ‘longevity’ or ‘longevity GWAS’ when referring to our analyses and results.

#7

Line 61-63: The authors state that “[proxy-based GWAS] may fail to comprehensively capture the genetic influences that directly impact an individual's lifespan” but the 2017 study by Liu et al. (PMID 28092683) which they cite does not necessarily support this argument. Liu et al. show that proxy-based GWAS effect estimates are directionally consistent with GWAS estimates of probands (albeit they need larger effective samples to achieve the same power). I acknowledge proxy-based GWAS may capture different genetic influences if the proxy samples are different (e.g. causes of death in the parental generation could be different from the current generation), but the authors should support this argument using more relevant citations.

(Response) One important result of our study is the influence of CHIP-related somatic variants on human lifespan in the UK Biobank. Such somatic variants association would be missed in a proxy-based approach, as exemplified on Fig.1 from Liu et al 2022 (PMID: 37117740) where solely the germline variants associations on *BRCA1* and *BRCA2* are identified in the proxy approach, while *TET2* and *ATM* associations are missed.

While CHIP susceptibility is linked to germline variants (<https://www.nature.com/articles/s41586-022-05448-9>), it is likely that genetic by environment interaction plays an important role in CHIP status in turn influencing lifespan. In that case, as emphasized by the reviewer, the proxy-based GWAS may perform even worse than on disease phenotypes reported by Liu et al 2017 (PMID 28092683).

We initially wrote:

“While proxy-based GWAS have been necessary for large cohorts [10], primarily of middle-aged individuals with limited mortality data, this approach restricts the accuracy and scope of findings, as it may fail to comprehensively capture the genetic influences that directly impact an individual's lifespan.”

[10] Liu, J.Z., Y. Erlich, and J.K. Pickrell, Case–control association mapping by proxy using family history of disease. *Nature Genetics*, 2017. 49(3): p. 325-331.

We have now rephrased this as:

“While proxy-based GWAS have been useful for gaining genomic insights into age-related diseases in cohorts primarily composed of middle-aged individuals, and show some consistency with associations related to lifespan (PMID: 37117740), they may fail to fully capture the genetic influences that directly impact individual lifespan, particularly CHIP-related somatic variants (PMID: 36450978).”

#8

The code provided by the authors is quite sparse and is missing the details of their phenotype quality control, the models used to fit Cox proportional hazard models of selected rare variants; and the entirety of the PheWAS. More detailed code would benefit the reproducibility of their study and the detail I can provide in my review.

To enhance the reproducibility of our analysis (fully described in the Methods section), we have added the suggested scripts and organized them into three folders (<https://github.com/Junkkkk/Lifespan-studies/tree/main>):

1. Scripts for calculating Martingale residuals.
 - i. A script is provided to define the last known age based on date of birth and the latest censoring date.
 - ii. The code for estimating Martingale residuals: These include updates to incorporate left truncation (Age_1st_visit), additional principal components (PC1-40), and geometric variables.
2. A Script for common variant GWAS using a BOLT-LMM model.
3. Rare variant analysis scripts.
 - i. A script on how to get annotation for rare variants.
 - ii. A script for SKAT-O and burden test
 - iii. A script with the models used for Cox proportional hazard modeling on selected rare variants.
 - iv. A script for generating Kaplan-Meier plots for significant genes identified in the burden test.
 - v. List of phenotypes analyzed in the PheWAS.

For the PheWAS, we employed the PHEnome Scan ANalysis Tool (PHESANT, PMID: 29040602, <https://github.com/MRCIEU/PHESANT>). Detailed methodology is provided in the referenced documentation, but we have uploaded a list of the 1,670 phenotypes analyzed in our study for reference.

#9

At the time of review (14 September 2024), PheWeb results for rs13190937 also include “Disorders of iron metabolism” ($P = 2.9 \times 10^{-60}$), which is worth mentioning given the suspected link between iron metabolism and morbidity/mortality (PMID: 30651232, 32678081). I recommend including the PheWeb lookup date in the main text for transparency.

Thank you for the insightful comment. PheWeb includes several databases. The one we referenced is 'UK Biobank Neale v1,' which is based on 337,000 unrelated white British individuals. The data reviewed by the Reviewer comes from the UK Biobank TOPMed-imputed dataset, which includes 400,000 white British individuals. We will clarify the database used in our manuscript to ensure readers know to which summary stats we refer to.

“Phenome-wide association study analysis (PheWAS) using PheWeb [15] based on ‘UKBiobank Neale v1 dataset’ shows that the main associations of rs13190937 are with celiac disease and intestinal malabsorption ($p=1.8 \times 10^{-57}$)”

#10

The code provided by the authors is quite sparse and is missing the details of their phenotype quality control, the models used to fit Cox proportional hazard models of selected rare variants; and the entirety of the PheWAS. More detailed code would benefit the reproducibility of their study and the detail I can provide in my review.

Please see our response to minor comment #8 above.

Title: Genetic associations with human lifespan are enriched for oncogenic genes

Responses to reviewers' comments

Please note that the revisions made have not influenced the content, conclusions, or framework of the paper. We have not listed all minor revisions made; however, these are highlighted in yellow in the revised manuscript.

Reviewer 1's comments

Thanks to the authors for responding to my comments. I am happy to recommend the paper for publication. I look forward to seeing it in print!

Reviewer #1 (Remarks on code availability):

I have not been through every line, but the repository is organised and I could find the information I was looking for. The authors should be commended for this!

Reviewer 2's comments

All my points were addressed, no further comments.

Reviewer 3's comments

I appreciate the authors' explanation as to why the Liu *et al.* study did not identify *ASXL1*, and commend their efforts to update their analyses and improve the reliability of their study. Also, the expansion of the code provided by the authors has allowed me to more closely examine the methods, which has been a great help.

My only remaining concerns are regarding the rare variant analysis and its replication, and the exclusion of the X chromosome in the common variant GWAS. Otherwise, most of my outstanding comments focus on improving the clarity and interpretation of the findings.

Major comments

#1

The use of BOLT-LMM has assuaged my concerns regarding statistical inflation due to participant relatedness in the common variant GWAS, but the authors make no mention of accounting for relatedness in the rare variant aggregation tests in the manuscript text and I am unable to find any such relatedness correction in their code. Previous studies which performed rare variant aggregation tests in UK Biobank invariably fit a genetic relatedness matrix (PMID: 36778668, 39210047) or remove related individuals (PMID: 31866045), so I would expect the authors to take family structure into account or justify why they chose not to account for it. I accept the genes highlighted by the authors plausibly affect lifespan through cancer incidence, but failure to account for relatedness could lead to overestimating the effects of these genes, and in turn, draw into question the authors' conclusion that preventative screening for such rare variants would have a material benefit. There are readily available tools like SAIGE-GENE+ and REGENIE which can perform rare variant aggregation tests in large cohorts while taking population structure and relatedness into account. Alternatively, the Martingale residuals phenotype itself could be adjusted for kinship using fastGWA's Gamma factor (--save-fastGWA-mlm-residual).

Thank you for raising this important point. To account for relatedness in the rare variant aggregation tests, we now adjusted the Martingale residuals phenotype for kinship using fastGWA's Gamma factor (--save-fastGWA-mlm-residual). This approach allowed us to efficiently account for relatedness while maintaining computational feasibility for such a large cohort.

Following this adjustment, we reanalyzed the rare variant associations and updated the corresponding figures and tables accordingly. Additionally, we have included the following description of this method in the Method section of the manuscript:

“To account for relatedness and population structure, Martingale residuals were first adjusted using a linear mixed model approach implemented in fastGWA (--save-fastGWA-mlm-residual) [31]. The adjusted residuals were then used as the phenotype for the rare variant analyses, ensuring the robustness of the results in the presence of related individuals and population stratification.”

#2

The authors use an independent UK Biobank sample to validate their rare variant results, but the genetic background of this validation sample consists of a mix of ancestries and, like the discovery sample, includes related individuals. Genetic associations are susceptible to population stratification, so ideally, each major genetic ancestry would be analysed separately and relatedness accounted for. I understand that rare variants require large samples to be testable, and splitting the validation sample by genetic ancestry reduces sample size, so I recommend at least one (but ideally both) of the following validation efforts:

##2.1

Fit major genetic ancestry groups and genetic relatedness as covariates when testing the current multi-ancestry replication sample. Only genes significant in the UK Biobank discovery sample need to be tested for replication, so the multiple testing threshold does not have to be too stringent ($P < 0.05 / \text{number of significant genes tested for replication}$). In reporting their results, the authors should distinguish which genes could be tested for replication, which genes replicate at nominal $P < 0.05$ significance, and which genes replicate at multiple testing corrected significance.

##2.2

Perform a leave-one-out cross-validation of the discovery dataset to check if the discovery effect estimates are stable. Specifically, the authors could leave out 20% of the British ancestry discovery sample and perform rare variant aggregation tests in the remaining 80% (accounting appropriately for relatedness as in my previous point). Repeating this analysis five times, where they exclude a different 20% of the sample in each iteration, should highlight which genes are stably associated with lifespan, and which genes are susceptible to population structure or bias in the discovery sample.

For the readers' clarity, I recommend the authors only mention replicated and/or cross-validated associations in the main text and include the full set of results in supplementary tables.

We appreciate your suggestion regarding handling multi-ancestry replication and performing five-fold cross-validation in our discovery dataset. We implemented both approaches as recommended. In response, we performed the following analyses:

1. **Multi-ancestry replication:** We excluded samples who self-reported as "Do not know" or "Prefer not to answer" for ethnicity, resulting in a final dataset of 73,281 individuals. These individuals were categorized into five groups based on their self-reported ethnicity (Field : 21000, <https://biobank.ndph.ox.ac.uk/showcase/field.cgi?id=21000>): White (66.3%), Asian (14.4%), Black (9.9%), Other (5.7%), and Mixed (3.7%). These groups were encoded as dummy variables and included as covariates in the calculation of Martingale residuals. Following this, we adjusted the Martingale residuals using fastGWA as described in # 1.

“In addition, we validated these findings within the UKB dataset using two approaches: an independent sample separate from our discovery data and a 5-fold cross-validation (CV) within the discovery cohort.

This independent validation included 73,281 subjects who were not categorized as having European ancestry based on genetic ethnic grouping. These participants were classified into five groups based on their self-reported ethnicity (Field: 21000): White (66.3%), Asian (14.4%), Black (9.9%), Other (5.7%), and Mixed (3.7%). Among the 21 novel genes identified in the discovery, four achieved significance under the Bonferroni correction threshold 1.1×10^{-3} ($0.05/42$) in this validation cohort: *BRCA2*, *ASXL1* with LoF variants and *IDH2*, *SRSF2* with pathogenic missense variants predicted by AlphaMissense (**Supplementary Table 4**).”

2. **Five-fold Cross-Validation:** Using 5-fold cross-validation, we divided our discovery dataset of 393,833 individuals into five folds. For each fold, 80% of the data (315,066 individuals) were used, and Martingale residuals were calculated using a Cox proportional hazards model that accounted for relatedness and population stratification. Rare variant aggregation tests were then performed on the datasets in five iterations, each time excluding a different 20% of the samples.

“To further validate our findings, we performed 5-fold CV within the discovery dataset of 393,833 individuals, dividing it into five folds. For each fold, 80% of the data (315,066 individuals) was used for analysis. The results across folds were highly consistent. For example, *TET2*, *BRCA2*, *BRCA1*, *ASXL1* (LoF), *SF3B1*, *DNMT3A*, *IDH2*, *TP53*, *SRSF2* (AlphaMissense) and *DNMT3A* (REVEL) achieved gene-wide significance across all five folds. Except for *CKMT1B* (LoF), *C1orf52*, *TET2*, *RLIM* (AlphaMissense) and *NMNAT2* (REVEL), all other genes that showed significance in the main analysis were significant in at least 3 out of 5 folds. This consistency across folds confirms the robustness of the associations identified in our study. Fold-specific results for each gene and variant category are provided in **Supplementary File 2**.”

#3

A closer look at the code suggests only autosomal chromosomes were tested for common variant GWAS associations and sex chromosomes were excluded, which is not explicitly stated in the main text. Excluding sex chromosomes is hard to justify when BOLT-LMM is able to test variants on the X chromosome (Y chromosome may be more complicated due to its non-recombining structure). For the rare variant analysis, the authors tested genes on the X chromosome, but I was unable to find the X chromosome-specific SKAT-O functions (SKAT_Null_Model_ChrX and SKAT_ChrX) in the author-provided code. Can the authors confirm the details of their rare variant X chromosome analysis?

We have added the analysis of chromosome X to the common variant GWAS using BOLT-LMM. Accordingly, we have updated the Manhattan plot and results in Figure 1. Additionally, it appears that the code for rare variant analysis on chromosome X was missing from our GitHub repository. We have now included the relevant code in the repository.

Minor comments

#1

The revised text would benefit from a careful check to remove any typos and to confirm consistency with the updated results (e.g. line 92 says “two variants were GWS” but goes on to discuss three variants; line 102 has an unnecessary apostrophe before UK Biobank, “FinnGen” is misspelt as “FinnGenn”, etc). Along the same lines, the heritability statement in the revised text differs from the statement mentioned in the rebuttal (Line 51-52). Please verify which is correct.

Thank you for highlighting these inaccuracies. We have addressed the specific issues you mentioned and thoroughly reviewed the text to make additional corrections and ensure overall consistency.

#2

There is a typo in the Lifespan-studies/1.Martingale_residual/Define_last_known_age.py script, where the censoring year is mistakenly set to 2022 instead of the year 2023 reported in the manuscript text.

Thank you for bringing this up. You are correct that the main text and the comment of code comment contain a typo. The correct censoring year is 2022, not 2023. As outlined in the UK Biobank documentation on data availability (https://biobank.ndph.ox.ac.uk/showcase/exinfo.cgi?src=Data_providers_and_dates), the censoring date for the most recent death data was November 30, 2022. Although the documentation notes that a small amount of additional data might be available beyond the censoring date, we chose to use the censoring date to ensure the highest level of accuracy. Accordingly, November 30, 2022, was used as the censoring date in our analysis.

[figure redacted]

#3

The author-provided code is missing sex-stratified versions of Martingale residual calculation, common variant GWAS, and rare variant burden analyses.

Yes, we have added the sex-stratified versions of the Martingale residual calculation, common variant GWAS, and rare variant burden analyses to our GitHub repository.

#4

The code in the Lifespan-studies/3.Rare_variants_studies/Cox_regression_rare_variants.R script contains an adjustment for cluster(FID), implying correction for broad family relatedness when testing rare variant carrier status, but this covariate is not mentioned in the method section. Also, the Cox proportional hazards model in this script does not take into account left truncation.

The inclusion of cluster(FID) in the code was not a new covariate but a typographical error. We have corrected it as follows and updated the script accordingly. Additionally, we have incorporated left truncation into the Cox proportional hazards model and reanalyzed the data.

```
“formula <- as.formula(paste("Surv(Age_1st_visit, last_known_age, Death) ~ Sex + ", paste(pc_columns, collapse = " + "), " + ", paste(c_columns, collapse = " + "), " + ", variant_type, "_carrier+cluster(FID)"))”
```

#5

Line 92-96: I believe the *ZSCAN23* SNPs rs111859903 and rs13190937 are almost in perfect linkage disequilibrium, so mentioning each as a separate association in the main text seems unnecessary. For clarity, it would be helpful to annotate the *ZSCAN23* locuszoom plot with all independent SNPs mentioned in the main text.

Reviewer #3 is correct that these genome-wide significant SNPs are almost in perfect linkage disequilibrium ($R^2 > 0.99$). Therefore, we mention only the top SNP based on p-value in the main text for clarity, and annotate all significant SNPs in the Figure 1D regional plot.

“On the chromosome 6 locus overlapping *ZSCAN23*, the top genome-wide significant variant is rs6902687, located 2.2 kb upstream of the transcription start site (TSS) (rs6902687_C: $\beta = 0.004$, $p = 2.7 \times 10^{-8}$, MAF = 36.6%). This variant is in almost perfect linkage disequilibrium ($R^2 > 0.99$) with three other significant variants in this region, including rs13215804_G (located 4.2 kb upstream of the TSS), rs111859903_G (located in an intron) and rs13190937_A (situated in the 5' untranslated region) (Figure 1D).”

#6

Line 122-128: It would be informative to mention the directionality in addition to significance when discussing replication of common variant results. For example, from my reading of the FinnGen data (<https://r11.finnngen.fi/variant/6:28443467-G-A>, accessed 6 November 2024), the direction of effect for the A allele of rs13190937 on 'Any death' is negative ($\beta = -0.011$, $p = 0.205$, i.e. increasing lifespan), while the effect on martingale residuals in the current study is positive ($\beta = +0.004$, $p = 2.9 \times 10^{-8}$, i.e. decreasing lifespan). In other words, the FinnGen association does not support the discovery. Note that it is also worth double-checking the numbers here as the trait name and P value I found differs from the one reported by the authors.

Thank you for pointing out this detail and confirming that the p-value for rs13190937 in FinnGen is indeed 0.20. As noted, rs13190937 was not significantly associated with the 'Any death' phenotype in FinnGen. Given that the p-value does not reach statistical significance, the directionality of the beta ($\beta = -0.011$) is not meaningful, and thus, the FinnGen dataset does not support our findings. We have updated our manuscript to reflect this and appreciate your attention to this detail.

“In FinnGen, rs13190937 was not significantly associated with the 'Death' phenotype ($p=0.2$), while it was significantly associated with a decrease in 'Parental age at death' in the UKB and LifeGen consortium ($p=1.4e-4$, $\beta=-0.015$).”

#7

The replication analysis is made even harder to interpret when looking at the LifeGen + UKB associations, described as 'Parental Martingale residuals' in the main text but as 'Parental age at death' in Supplementary Table 2. The authors report a positive effect in the table for rs13190937 ($\beta = 0.015$, $P = 1.4 \times 10^{-4}$), suggesting a lifespan-increasing effect if the phenotype is age at death, which would contradict the discovery. More clarity in the text and table regarding directionality would greatly aid the reader.

Thank you for your insightful comment. First, we acknowledge that the replication analysis phenotype is indeed "Parental age at death", and we have updated the main text accordingly to ensure consistency. Additionally, the beta sign for the effect allele (A) is -0.015, not positive as previously mentioned.

To clarify further, the figure below shows a portion of the summary statistics from the study by Timmers, P.R., et al., which we referenced, and can be accessed in full at https://ftp.ebi.ac.uk/pub/databases/gwas/summary_statistics/GCST009001-GCST010000/GCST009890/.

rsid	snpid	chr	pos	a1	a0	n	freq1	beta1	se	p
rs35705950	11_1241221	11	1241221	T	G	583397	0.11427378	-0.0231996	0.00662653	0.0004635
rs13190937	6_28411244	6	28411244	A	G	640183	0.35273204	-0.0153619	0.0040266	0.00013612
rs429358	19_4541194	19	4541194	T	C	626548	0.84550672	0.1056076	0.00546403	3.14E-83

We have also included beta directions for significant variants in the replication data in the main text for greater clarity.

“In FinnGen, rs13190937 was not significantly associated with the 'Death' phenotype ($p=0.2$), while it was significantly associated with a decrease in 'Parental age at death' in the UKB and LifeGen consortium ($p=1.4e-4$, $\beta=-0.015$).”

#8

In line with the previous point, reporting directionality for the PheWAS would be useful as well. For example, is the lifespan-reducing allele of rs13190937 associated with increased or decreased celiac disease? One would assume the former but it would be helpful to explicitly report the effect size and direction.

In the PheWAS results, we confirmed that rs13190937 is associated with an increased risk of celiac disease with an OR of 1.003 ($p = 1e-57$). We have updated the main text to include both the p-value and the OR for clarity.

#9

In line with Reviewer #1's comment on "novelty", figure legends would benefit from a more detailed description of what "novelty" means exactly. In addition, explicitly stating the sample sizes and significance thresholds in figure legends would make it easier for the reader to interpret the results.

To enhance clarity for the reader, we have added a detailed explanation of the meaning of "novel genes" to the legends of Table 1 and Figure 2, along with the significance thresholds. Additionally, for Figure 3, we have clarified the meaning of the numbers shown in the figure within the legend.

“Table 1. Significant genes for rare variants association with burden and SKAT-O tests ($p < 7.4 \times 10^{-7}$). Genes names in bold font represent those not previously identified as significant in [8].”

“Figure 2. Rare variant burden association with lifespan, considering loss-of-functions (A) and Alpha Missense pathogenic variants (B). Genes highlighted in red represent those not previously identified as significant in [8]. A gene-wide significance threshold of $p = 7.4 \times 10^{-7}$ was applied.”

“Figure 3. Survival curves comparing carriers and non-carriers of variants on genes with a significant burden of loss-of-function (A) and AlphaMissense pathogenic (B) variants. For each gene, the survival curve includes Cox regression hazard ratio (HR), p-value, the number of carriers, and their proportion within the total sample. A gene-wide significance threshold of $p = 7.4 \times 10^{-7}$ was applied.”

#10

In Supplementary Figure 1 and 2E, certain phenotypes have been underlined, but it is unclear from the legend why these phenotypes were highlighted.

Supplementary Figure 1 presents the PheWAS results for rs13190937 in *ZSCAN23*. Colocalization analysis indicated an association between rs13190937 and pancreatic tissue. Diseases potentially related to pancreatic tissue among the significant PheWAS results were highlighted. Similarly, Supplementary Figure 2E shows the PheWAS results for rs35705950, which colocalized with lung tissue. Significant diseases related to lung conditions were highlighted in the Supplementary Figure 2E. However, as no strong evidence of deeper connections could be established, we have removed the highlights for clarity.

#11

There is no explanation in the Table 2 legend regarding the content of the "Reported" column: are these studies reporting associations with the gene or the variant? Are you allowing for proxy variants in high linkage disequilibrium? Moreover, it is unclear if the list of reported phenotypes is complete: the authors themselves show in their PheWAS analysis that many of the variants are associated with one or more phenotypes in Neale's analysis of UK Biobank (e.g. *CKMT1B* LoF and Hypopharynx cancer).

Table 2 "Reported" column lists published studies that previously reported the highlighted variants as associated with a given phenotype (Clinvar search, <https://www.ncbi.nlm.nih.gov/clinvar/>). While Neale's analysis is publicly available, the particular variants (especially rare variants) are not referenced in the companion study. We report Neale's group associations as part of our PheWas analysis query in the UKB. We clarify the meaning of this column in the Table 2 legend.

"Table 2. Lead variant association per gene among significant genes in the burden and SKAT-O tests. Only variants with at least 3 minor alleles are reported. A significance threshold of $p=8.3 \times 10^{-5}$ was applied after a Bonferroni correction for multiple testing. The "Reported" column indicates published studies that associated the highlighted variants with specific diseases, curated from ClinVar (<https://www.ncbi.nlm.nih.gov/clinvar/>)."

#12

I suggest the authors edit the manuscript title to more accurately reflect the type of variants and the sample cohort used to identify oncogenic genes, as the current title appears to make the broad claim that all lifespan variants across all populations are enriched for oncogenic genes, while the study only identifies this link among rare variant associations in UK Biobank. One suggestion would be "Rare genetic associations with human lifespan in UK Biobank are enriched for oncogenic genes".

Thank you for the suggestion. We agree with your feedback and have revised the manuscript title accordingly to:

"Rare genetic associations with human lifespan in UK Biobank are enriched for oncogenic genes."

#13

The discussion focuses largely on the findings but misses an opportunity to discuss the limitations of the study. For one, only a small number of variant and gene burden associations could be formally replicated, but this lack of replication is not acknowledged and the possible reasons for it are not discussed. Similarly, around line 392 the discussion would be enriched by including the authors' explanation of *ASXL1* (as mentioned in their rebuttal), with the added takeaway that the selection of variants can strongly influence the findings, and is therefore an important limitation of the study.

Thank you for this suggestion. We now highlighted the lack of independent replication as one of the limitations of our study in the discussion:

“While our study lacks an independent external replication and only a small number of genes formally replicated in the independent test set within UKB, the stability of the burden test association to five-fold CV suggests that these results are not due to outliers and are robust within the UKB.”

#14

Line 385-389: Which previous lifespan associations were checked? There is no mention of this analysis in the main text or any table to support the statement. Additionally, the authors state the phenomenon likely resulted from proxy vs. proband data, but an alternative explanation could be the differences in sample size. A quick calculation of the sample size needed to detect the previously reported effects could help determine to what degree sample size plays a role in the discrepancy.

“Previously reported SNP associations with lifespan were concordant in our dataset but none of these passed the GWAS suggestive threshold ($p=1.0e-5$) except for those at the *APOE* locus (**Supplementary Table 9**).”

“**Supplementary Table 9. Previously reported SNP associations with lifespan from prior studies (excluding the *APOE* region).** This table summarizes SNPs previously associated with lifespan-related phenotypes in published studies that were GWAS significant in their respective studies. Notably, none of the SNPs in this table reached the suggestive threshold ($p=1.0 \times 10^{-5}$) in our analysis.”

Regarding our interpretation of non-replication,

“This phenomenon likely resulted from previous studies relying on proxy data such as parental age at death, which may capture a different set of genetic factors than direct proband mortality data.”

The previous effective sample sizes are smaller than the current sample size of directly reported death in our study. Thus, we believe that we were adequately powered to replicate these associations. In addition, different periods of times or geographical locations likely have different gene by environment interactions leading to different lifespans, and thus some genetic associations identified with parental proxy may not be re-produced in offspring.

#15

Please provide a Supplementary Data file with the list of SNPs for each gene used to calculate burden and SKAT-O statistics. This will allow other researchers to more easily reproduce the authors’ analysis.

We agree and have added a list of SNPs classified by loss-of-function, AlphaMissense, and REVEL for each gene to the **Supplementary File 1**. This addition will allow other researchers to more easily reproduce our analysis.